

# Modelling $CO_2$ weather – why horizontal resolution matters

Anna Agustí-Panareda[1], Michail Diamantakis[1], Sébastien Massart[1], Frédéric Chevallier[2],
Joaquín Muñoz-Sabater[1], Jérôme Barré[1], Roger Curcoll[3], Richard Engelen[1], Bavo Langerock[4],
Rachel Law[5], Zoë Loh[5], Josep Anton Morguí[3], Mark Parrington[1], Vincent-Henri Peuch[1],
Michel Ramonet[2], Coleen Roehl[6], Alex T. Vermeulen[7], Thorsten Warneke[8], and Debra Wunch[9]

[1]European Centre for Medium-range Weather Forecasts, Shinfield Park, Reading, RG2 9AX, United Kingdom
[2]Laboratoire des Sciences du Climat et de l'Environnement, LSCE/IPSL, CEA-CNRS-UVSQ, Université Paris-Saclay,
F-91198 Gif-sur-Yvette, France
[3]Environmental Science and Technology Institute, Universitat Autònoma de Barcelona, ICTA-UAB, Bellaterra, Spain
[4]Royal Belgian Institute for Space Aeronomy (BIRA-IASB), Brussels, Belgium
[5]CSIRO Oceans and Atmosphere, PMB 1, Aspendale, Victoria 3195, Australia
[6]California Institute of Technology, Pasadena, California, USA
[7]ICOS ERIC- Carbon Portal, Sölvegatan 12, 22362 Lund, Sweden
[8]University of Bremen, Institute of Environmental Physics, Otto-Hahn-Allee 1, 28359 Bremen, Germany
[9]University of Toronto, Department of Physics, Toronto, Ontario, Canada

**Correspondence:** Anna Agusti-Panareda (Anna.Agusti-Panareda@ecmwf.int)

**Abstract.** Climate change mitigation efforts require information on the current greenhouse gas atmospheric concentrations and their sources and sinks. Carbon dioxide ($CO_2$) is the most abundant anthropogenic greenhouse gas. Its variability in the atmosphere is modulated by the synergy between weather and $CO_2$ surface fluxes, often referred to as $CO_2$ weather. It is interpreted with the help of global or regional numerical transport models, with horizontal resolutions ranging from

5  a few hundreds of km to a few km. Changes in the model horizontal resolution affect not only atmospheric transport, but also the representation of topography and surface $CO_2$ fluxes. This paper assesses the impact of horizontal resolution on the simulated atmospheric $CO_2$ variability with a numerical weather prediction model. The simulations are performed using the Copernicus Atmosphere Monitoring Service (CAMS) $CO_2$ forecasting system at different resolutions from 9 km to 80 km and are evaluated using in situ atmospheric surface measurements and atmospheric column-mean observations of $CO_2$, as well as

10  radiosonde and SYNOP observations of the winds.

The results indicate that both diurnal and day-to-day variability of atmospheric $CO_2$ are generally better represented at high resolution, as shown by a reduction in the errors in simulated wind and $CO_2$. Mountain stations display the largest improvements at high resolution as they directly benefit from the more realistic orography. In addition, the $CO_2$ spatial gradients are generally improved with increasing resolution for both stations near the surface and those observing the total column, as the

15  overall inter-station error is also reduced in magnitude. However, close to emission hotspots, the high resolution can also lead to a deterioration of the simulation skill, highlighting uncertainties in the high resolution fluxes that are more diffuse at lower resolutions.

We conclude that increasing horizontal resolution matters for modelling $CO_2$ weather because it has the potential to bring together improvements in the surface representation of both winds and $CO_2$ fluxes, as well as an expected reduction in numer-



ical errors of transport. Modelling applications like atmospheric inversion systems to estimate surface fluxes will only be able to benefit fully from upgrades in horizontal resolution if the topography, winds and prior flux distribution are also upgraded accordingly. It is clear from the results that an additional increase in resolution might reduce errors even further. However, the horizontal resolution sensitivity tests indicate that the change in the $CO_2$ and wind modelling error with resolution is not linear,

making it difficult to extrapolate the results beyond the tested resolutions.

Finally, we show that the high resolution simulations are useful for the assessment of the small-scale variability of $CO_2$ which cannot be represented in coarser resolution models. These representativeness errors need to be considered when assimilating in situ data and high resolution satellite data such as Greenhouse gases Observing Satellite (GOSAT), Orbiting Carbon Observatory-2 (OCO-2), the Chinese Carbon Dioxide Observation Satellite Mission (TanSat) and future missions such as the

Geostationary Carbon Observatory (GeoCarb) and the Sentinel satellite constellation for $CO_2$. For these reasons, the high resolution $CO_2$ simulations provided by the CAMS in real-time can be useful to estimate such small-scale variability in real time, as well as providing boundary conditions for regional modelling studies and supporting field experiments.

## 1 Introduction

Over synoptic weather time scales of hours to days and spatial scales less than $1000\,\mathrm{km}$, the assumption that atmospheric $CO_2$

is well-mixed into a homogeneous background does not hold, as shown by the observed variability at baseline in situ stations (e.g. Halter et al., 1983). $CO_2$ weather is defined here as the atmospheric $CO_2$ variability at timescales and spatial scales of weather systems (Parazoo et al., 2011) as depicted in Fig. 1. It reflects a complex combination of anthropogenic and natural $CO_2$ fluxes near the Earth's surface and transport by weather systems in the atmosphere (Geels et al., 2004; Patra et. al, 2008). This synergy of $CO_2$ fluxes and weather results in intricate atmospheric $CO_2$ patterns of positive and negative anomalies,

collocated with weather variations on top of the well-mixed $CO_2$ background that varies slowly on timescales of weeks to annual timescales (Keeling et al., 1976).

Modelling the synoptic-scale transport that modulates the $CO_2$ weather is crucial for interpreting the variability of surface $CO_2$ concentrations from in situ observations (Law et al., 2010) and column-averaged $CO_2$ from satellite and ground-based observations (Corbin et al., 2008), and for forecasting $CO_2$ from 1 to 10 days ahead (Agustí-Panareda et al., 2014; Tang et al.,

2018) in order to examine the predictive skill of the models. Tracer transport models use the numerical schemes and meteorological information of Numerical Weather Prediction (NWP) to simulate the tracer variability in the atmosphere. Increasing the horizontal resolution of tracer transport models has the benefit of reducing the numerical errors in tracer simulations, leading to convergence of the transport solution from different transport schemes (Prather et al., 2008). NWP models for weather forecast have been doubling the global horizontal resolution approximately every 8 years (Wedi, 2014) in order to improve the forecast

skill. But until now, global tracer transport models generally use lower resolution than NWP models, as chemical transport models including chemistry and/or long window data assimilation cannot afford such computational expense.

Observations of atmospheric $CO_2$ are used in data assimilation systems based on tracer transport models to produce optimal estimates of atmospheric $CO_2$ concentrations (e.g. Massart et al., 2016) or model parameters and $CO_2$ fluxes in atmospheric



inversion systems (e.g. Rayner et al., 2005; Chevallier et al., 2010). If tracer transport models cannot represent the synoptic variability accurately, then the resulting errors when comparing the tracer from the model with observations will prevent these observations to be used effectively in the data assimilation systems (e.g. Brooks et al., 2012). The model-observation mismatch caused by differences in the resolution of the tracer transport model – including both the resolution of the meteorological

fields and the resolution of the fluxes on the model grid – and the resolution of the observation footprint is also known as representativeness error. Failure to properly account for representativeness errors in data assimilation will lead to errors in the optimized parameters, being atmospheric concentrations, model parameters or surface fluxes.

Several studies have investigated the spatial representativeness errors of $CO_2$ (Miller et al., 2007; van der Molen and Dolman, 2007; Corbin et al., 2008; Tolk et al., 2008) by analysing the $CO_2$ distribution within model grid cells, based on nested high

resolution simulations on limited domains over Europe, North America and South America for certain months or by statistical parameterization of $CO_2$ covariances based on lower resolution simulations (Alkhaled et al., 2008). The importance of high resolution over complex terrain has also been demonstrated on regional scales, e.g. in Europe (van der Molen and Dolman, 2007; Ahmadov et al., 2009; Pillai et al., 2011) and in North America (Lin et al., 2017; Hedelius et al., 2017) using very high resolution simulations (down to 1 km). However, other studies with coarser global tracer transport models have compared $CO_2$

simulations with a range of resolutions from a few degrees down to 0.5 degree without finding significant improvements with respect to observations (Lin et al., 2018; Remaud et al., 2018).

The full impact of horizontal resolution on the simulated tracer variability depends on the resolution of transport and emissions/biogenic fluxes (e.g. Vogel et al., 2013) as well as the resolution of the topography and the winds (e.g. Sekiyama et al., 2015) in the model. In this study the full sensitivity of $CO_2$ synoptic variability to the model horizontal resolution (including all

the aspects mentioned above) is investigated by quantifying the change in model error with horizontal resolution at observing stations. Three main questions are addressed:

1. What is the sensitivity of the modelled atmospheric $CO_2$ variability at diurnal and synoptic timescales to horizontal resolution?

2. How is horizontal resolution affecting the medium-range (1-10 day) forecast error growth of atmospheric $CO_2$?

3. What are the typical $CO_2$ representativeness errors in models with horizontal resolutions of 1 degree x 1 degree, currently considered as high resolution in tracer transport models; and where and when are these representativeness errors largest?

The model simulations use the operational Copernicus Atmosphere Monitoring Service (CAMS) global $CO_2$ forecasting system (Agustí-Panareda et al., 2014; Massart et al., 2016) which is based on the Integrated Forecasting System (IFS) model of the European Centre for Medium Range Weather Forecasts (ECMWF). They are performed over a range of resolutions

currently used operationally in NWP from 9 to 80 km. A detailed description of the simulations, observations and tools used to assess the importance of horizontal resolution for simulating atmospheric $CO_2$ variability related to weather is presented in section 2. Section 3 shows the impact of horizontal resolution on the error of simulated horizontal winds (section 3.1) and atmospheric $CO_2$ (sections 3.2 and 3.3). The results of the sensitivity to horizontal resolution are explained in the context





of the small-scale variability in section 3.4. The diagnostics of small-scale variability provide an estimate of the expected representativeness errors for $CO_2$ simulations with coarser horizontal resolutions. Finally, an example of an urban site is shown in section 3.5, where the impact of horizontal resolution is positive in January and negative in July. The implications of the results for $CO_2$ forecasting and atmospheric inversion systems are discussed in section 4 with a summary of the main
findings on why and where horizontal resolution matters.

## 2 Methodology

### 2.1 Observations

Continuous in situ observations near the surface and column-average observations from the Total Carbon Column Observing Network (TCCON) provide the reference for atmospheric $CO_2$ variability. Figure 3 shows the spatial distribution of the $CO_2$
observing stations used in this study. Hourly near-surface $CO_2$ observations are provided by 51 in situ stations operated by various organizations throughtout the period of the simulations: data from 44 stations are taken from the cooperative GLOB-ALVIEWplus ObsPack  (2015) data set, and additional data has been obtained from 3 additional stations from CSIRO in Australia and Antarctica, and 4 stations from the ClimaDat network (Morguí et al., 2013, 2017) over the Iberian peninsula. The cooperative GLOBALVIEWplus ObsPack  (2015) data set is coordinated by NOAA and it comprises data collected by various
institutions and laboratories including AEMET, AGH, CSIRO, ECCC, ECN, EMPA,FMI, HMS, LSCE, NCAR, NOAA, JMA, NIWA, SAWS, TU, UBA-SCHAU, UEA, UHEI-IUP and UR (see Tables A1 and A3 for full list of stations with their organisations and associated references). No selection criteria are applied to the stations from the GLOBALVIEWplus ObsPack (2015), CSIRO and ClimaDat datasets, other than availability of hourly data for January and July 2014.

Most stations are on the World Meteorological Organization (WMO) $CO_2$ scale, although the inter-calibration of standard
gases is not critical for this study because the focus is on the relative difference between the high and low resolution simulations to quantify the sensitivity of modelled $CO_2$ to horizontal resolution in the model. The distribution of the stations is not homogeneous over the globe. However, there is a wide variety of locations that sample synoptic variability on various types of terrain including many coastal, mountain, continental and oceanic sites over different continents on both hemispheres. Wind observations from around 400 radiosondes stations and all the operational 10-m SYNOP stations around the globe are used to
evaluate the sensitivity of wind errors to the model horizontal resolution at different atmospheric levels in the troposphere.

Total column observations from 18 TCCON Fourier Transform Spectrometer (FTS) network (Wunch et al., 2011) available in January 2014 and July 2014 – shown as red triangles in Fig. 3 – are also used to evaluate the variability of the column-averaged dry-air mole fraction of $CO_2$ – hereafter referred to as $XCO_2$ – (Tab. A2). These TCCON observations are retrieved from direct solar near-infrared spectra (www.tccon.caltech.edu) and they provide a ground reference to the GOSAT (Kuze et al.,
2009), OCO-2 (Crisp et al., 2017) and TanSat (Yang et al., 2018) satellite observations (e.g. Inoue et al., 2016; Wunch et al., 2017). Total column averages are less sensitive to the uncertainties associated with vertical mixing than the $CO_2$ abundances near the surface. However, the temporal coverage of TCCON observations is limited to clear sky and sunny conditions, which means there are generally more gaps in the TCCON data than in near-surface in situ data.



## 2.2 Global atmospheric $CO_2$ model

The model used in this study is the Integrated Forecasting System (IFS), the same model used in NWP at ECMWF and in the CAMS atmospheric composition analysis and forecasting system to issue 5-day $CO_2$ and $CH_4$ forecasts (https://atmosphere. copernicus.eu/maps/global-carbon-dioxide-forecast), as well as reactive gases and aerosol forecasts relevant for air quality
(Flemming et al., 2015; Morcrette et al., 2009). The IFS model version is CY43r1, the operational weather forecast model at ECMWF from 22 November 2016 to 10 July 2017. A full evaluation of this model cycle can be found in Haiden et al. (2017). The high horizontal resolution is based on a cubic octahedral reduced Gaussian – called hereafter octahedral – grid (Holm et al., 2016). The implementation of the octahedral grid has allowed a substantial increase in the grid point resolution from 16 km to approximately 9 km, without having to increase the spectral resolution of the model (Malardel et al., 2016). The 9 km
simulation comprises up to 904 million model grid points, 137 levels and a time step of 7.5 minutes.

The tracer transport is modelled by three different numerical schemes to represent (i) the resolved advection of $CO_2$ by the winds, and the sub-grid scale (ii) convection and (ii) turbulent mixing processes that need to be parametrized. The tracer advection is computed by a semi-implicit semi-Lagrangian scheme (Temperton et al., 2001; Diamantakis and Magnusson, 2016) which is an unconditionally stable method for the integration of the transport equations and for the fast terms associated with gravity waves. Semi-Lagrangian advection schemes have small dispersion and phase speed errors despite using long
timesteps (Staniforth and Côté, 1991). In practice, these properties mean that the timestep is limited only by the local truncation error and not by numerical stability bounds. The semi-lagrangian advection scheme in the IFS is not mass conserving. Thus, a mass fixer is required to ensure mass conservation at every time step (Agustí-Panareda et al., 2017; Diamantakis and Agustí-Panareda, 2017). The turbulent mixing scheme is described in Beljaars and Viterbo (1998); Koehler et al. (2011); Sandu
et al. (2013). The convection scheme is based on Tiedtke (1989) (see Bechtold et al., 2008, 2014, for further details ). Full documentation of the IFS can be found in https://www.ecmwf.int/en/forecasts/documentation-and-support/changes-ecmwf-model/ifs-documentation.

The $CO_2$ surface fluxes from the ocean, biomass burning and anthropogenic emissions are prescribed using inventories or climatologies, while the biogenic fluxes over land are modelled on-line (see Table 1). The anthropogenic $CO_2$ emissions come
from the EDGAR v4.2FT2010 inventory for 2012 (last year with gridded emissions). They are extrapolated in time to the year of the simulation with country trends provided by the EDGAR database (http://edgar.jrc.ec.europa.eu). The biogenic $CO_2$ emissions from land vegetation are modelled with the A-gs photosynthesis scheme and an empirical model to simulate the ecosystem respiration fluxes which are integrated in the CHTESSEL land surface model of the IFS (Boussetta et al., 2013). The fluxes have been evaluated with FLUXNET data and compared to different models (e.g. CASA and ORCHIDEE) showing
a comparable performance on synoptic to seasonal scales (Balzarolo et al., 2014). An on-line bias correction scheme (Agustí-Panareda et al., 2016) is applied to the modelled Gross Primary Production (GPP) and ecosystem respiration (Reco) fluxes to correct for biases in the NEE budget on a time-scale of 10 days compared to a climatology of optimized fluxes (Chevallier et al., 2010).





The atmospheric tracer transport and $CO_2$ biogenic fluxes are two of the largest contributors to the synoptic variability of atmospheric $CO_2$ globally (Geels et al., 2004; Agustí-Panareda et al., 2014). Thus, the modelling of these two components on-line in the IFS allows to investigate the full impact of the resolution coming from the winds, the tracer transport, as well as the fluxes.

## 2.3 Global atmospheric $CO_2$ simulations

A set of global simulations are performed at several resolutions from 9 km to 80 km (Table 2) to investigate the impact of horizontal resolution on the modelled $CO_2$ variability at diurnal and synoptic scales. These are the resolutions that are currently used operationally in global meteorological re-analysis – e.g. ERA-Interim at 80 km (Dee et al., 2011) - widely used in tracer transport models, and the typically higher resolutions of operational weather forecasts models. For instance, the deterministic weather forecast at ECMWF currently runs at 9 km resolution, and it was the global forecasting system with the highest resolution in the world when it was introduced on 8 March 2016 (Holm et al., 2016).

The octahedral grid is used for all simulations, except for the lowest resolution simulation at 80 km which uses a reduced linear Gaussian grid as in the ERA-Interim and CAMS re-analysis (Inness et al., 2018). The time steps are also dependent on the horizontal resolution and range from 7.5 minutes to 45 minutes. As described in section 2.2, the Semi-Implicit Semi-Langrangian method used in the IFS is free from stability restrictions. Thus, the model uses the longest possible timestep that provides the most accurate result for each spatial resolution. This is selected through experimentation and validation, but a rule of thumb is that as the horizontal resolution increases, the timestep decreases to keep the mean Courant–Friedrichs–Lewy (CFL) number constant. This typically leads to much longer timesteps than Eulerian models for which their timestep is restricted by the typical CFL stability limit (i.e. the maximum CFL number being less than 1).

All the simulation experiments are conducted for a winter and a summer month, in January 2014 and July 2014, as we expect that winter and summer periods will show markedly different variability patterns in $CO_2$. Figure 2 shows the configuration of the simulations. A 10-day forecast is performed at 00 UTC each day of the month. The meteorological initial conditions of each forecast come from the ECMWF operational NWP analysis (Rabier et al. , 2000); whereas the atmospheric $CO_2$ tracer is initialised with the previous 1-day forecast, which means $CO_2$ is essentially free running, as in Agustí-Panareda et al. (2014). The first initial conditions for $CO_2$ on 1 January 2014 and 1 July 2014 are extracted from the CAMS $CO_2$ analysis (Massart et al., 2016). NWP analysis of meteorological fields is one of the main elements determining the quality of the tracer transport (Locatelli et al., 2013; Polavarapu et al., 2016). Keeping the meteorological fields close to the analysis by having a sequence of 1-day forecasts ensures the tracer transport is as realistic as possible. Therefore, the sequence of 1-day forecasts is used as the standard (cyclic forecast) configuration for the simulations at different resolutions.

The extension to the 10-day forecasts allows to assess the impact of errors in the meteorological fields – which grow during the forecast – on the $CO_2$ simulations. There are 10 realisations of $CO_2$ for each day, one for each forecast lead time (Fig. 2). Each forecast lead time is evaluated separately in order to estimate the error growth during the forecast. For consistency in the evaluation of the different forecast lead times, the periods from 10 January to 10 February and 10 July to 10 August are used in the validation diagnostics.





The simulations also include an additional $CO_2$ tracer which is only transported (i.e. does not respond to $CO_2$ surface fluxes) during the forecast. We refer to this tracer as $NFX$. This tracer is still initialiased with the standard $CO_2$ at the beginning of each forecast. The difference between the $NFX$ $CO_2$ and the standard $CO_2$ tracers can provide insight on the sensitivity to local flux at different horizontal resolutions. Similarly, the change in the error of the simulation with resolution for both the standard and the $NFX$ tracers can be used as an indicator of transport versus local flux influence in the assessment of the impact of horizontal resolution.

## 2.4 Diagnostics for model evaluation

The focus of this paper is on assessing the skill of the model in simulating $CO_2$ weather with short-term variability over a period of a month. For this purpose, the Root Mean Square Error

$$RMSE = \sqrt{\frac{1}{N} \sum_{i=1}^{N} (m_i - o_i)^2}, \tag{1}$$

the systematic error or bias

$$\mu = \frac{1}{N} \sum_{i=1}^{N} (m_i - o_i) \tag{2}$$

and the random error

$$STDE = \sqrt{\frac{1}{N} \sum_{i=1}^{N} (m_i - o_i - \mu)^2} \tag{3}$$

of the modelled $CO_2$ dry molar fraction ($m$) are computed with respect to $N$ hourly observations ($o$) at each observing site. The standard deviation of the site error – also known as inter-station error – is used as an indicator of the spatial variability of the error $e$ (e.g. RMSE, $\mu$) between the $M$ observing sites:

$$\sigma_e = \sqrt{\frac{1}{M-1} \sum_{s=1}^{M} (e_s - \bar{e})^2}, \tag{4}$$

where $\bar{e}$ is the mean error of all sites. It reflects the skill of the model in representing spatial gradients between the sites. The Pearson's correlation coefficient is also used to assess the skill of the model in simulating the diurnal and synoptic variability at the sites.

The model is sampled in the horizontal by taking the nearest grid point to the station over land. This approach is widely used in model evaluation (Patra et. al, 2008) as it allows to assess the model directly at grid point scale. At coastal locations, coarse resolution models can find a better fit to observations by sampling the nearest ocean grid point as land grid points tend to overestimate the diurnal cycle (Law et al., 2010). For this reason, the sampling protocol for observations in atmospheric inversion system move some stations offshore (Gurney et al., 2003). However, coastal sites can have both ocean and land influence which means that they will have contrasting periods sampling baseline air associated with low variability and periods





with land and local influences associated with high $CO_2$ variability (Patra et. al, 2008). In this study we have chosen to sample consistently the nearest land point over land because we are interested in assessing the capabilities of the model to represent both baseline and local influences. The temporal sampling is performed with a linear interpolation from the 3-hourly archived model fields to the observation time as in Agustí-Panareda et al. (2014).

At the surface stations, the model is also interpolated to the altitude of the sampling height above the ground level (AGL). This ensures the same model levels are used for the different horizontal resolutions. The model has hybrid coordinates that follow the terrain close to the surface. Selecting the model level at the station height above mean sea level (ASL) would imply using different model levels for different resolutions when the orographic height varies between the horizontal resolutions. It would therefore lead to comparisons of $CO_2$ in the planetary boundary layer and free troposphere at mountain sites where the

low resolution model underestimates the orographic height. Lin et al. (2017) tested both approaches at several mountain sites. They found that the sampling at ASL greatly underestimates the amplitude of the diurnal cycle, as the sensitivity to local fluxes is reduced at higher levels above the ground. Since most low resolution models used in atmospheric inversions tend to use the model sampling ASL at mountain sites, a comparison of the two approaches (AGL and ASL) is provided in the supplement. At the TCCON stations, the model profile is processed with the TCCON averaging kernel and prior, as described in Appendix

A of Massart et al. (2016).

    Atmospheric $CO_2$ variability is subject to local or small-scale influences ($< 100\,km$) associated with complex topography, coastal boundaries, local fluxes and mesoscale atmospheric flow (Lin, 2007). Most models used in carbon cycle studies are unable to represent such local variability. The resulting representativeness errors reflect the sub-grid scale variability associated with the coarse resolution of the models (e.g. Tolk et al., 2008). At high resolution it is possible to estimate part of this

sub-grid scale variability of coarser models. In order to do that, the $9\,km$ simulation is interpolated onto a 0.1x0.1 degree regular lat/lon grid and subsequently it is sampled for each time zone (computed hourly along longitude) at 13:00 +/- 0.5 hours local time. This temporal sampling at 13:00 is consistent with the GOSAT (http://www.gosat.nies.go.jp/en/ and OCO-2 (http://oco.jpl.nasa.gov/mission) overpass time. Thus, it provides a more relevant estimate of the potential representativeness error for lower resolution inversion systems, which use daytime surface in situ data and satellite data (e.g. Chevallier et al.,

2010, 2014). The representativeness error is estimated by computing the standard deviation of the $CO_2$ dry molar fraction at 0.1 degree resolution within the coarser grid boxes of 1 degree x 1degree over the whole globe:

$$\sigma_{CO2} = \sqrt{\frac{1}{n-1}\sum_{j=1}^{n}(m_j - \overline{m})^2} \tag{5}$$

where $\overline{m} = \frac{1}{n}\sum_{j=1}^{n}m_j$; $n$ is the number of 0.1 degree resolution grid cells within the coarser grid cell of 1 degree x 1 degree; $m$ is the $CO_2$ dry molar fraction at 0.1 degree resolution; and $\overline{m}$ is the average within the coarser grid cell.





## 3 Results

### 3.1 Impact of horizontal resolution on winds

The accuracy of the winds is a crucial aspect of the $CO_2$ transport quality, as winds drive the advection of $CO_2$ across the resolved gradients in the model. In this section we investigate the benefit of increasing the resolution from 80 km to 9 km
on RMSE of the zonal and meridional components of the wind. We investigate the changes in the global wind error with model resolution based on 12-hourly radiosonde observations which measure the horizontal wind components throughout the troposphere. Figure 4 shows there is a consistent and signficant RMSE reduction of the vector wind RMSE for the 1-day forecast with resolution. The impact of resolution – quantified here by the difference in RMSE between the 80km and 9km simulations – is largest near the surface at 850 hPa and 1000hPa with a RMSE reduction ranging between 0.2 m/s and 0.6 m/s.
This is equivalent to a reduction in RMSE of around 15% near the surface. In the mid and upper troposphere (500 hPa and 200 hPa) there is a consistent but smaller RMSE reduction, ranging between 0.1 m/s and 0.2 m/s.

    The RMSE reduction extends throughout the 10-day forecast for the two components of the wind between 1000 and 850 hPa with values around 0.4 m/s and it is consistent in both northern/southern hemispheres and tropics (not shown). The results are also in agreement with the RMSE with respect to 10m wind speed from SYNOP observations with a mean RMSE reduction
over the global domain of 0.34 m/s. The reduction of the mean error is smaller than the RMSE (< 0.2 m/s) throughout the troposphere, which means the largest component of the wind error is random.

### 3.2 Impact of horizontal resolution on $CO_2$ diurnal and synoptic variability

The sensitivity of the model skill at hourly and daily time scales to the horizontal resolution of the model is assessed with the error of the $CO_2$ simulations with respect to hourly mean observations. The change in the RMSE with horizontal resolution
based on the surface $CO_2$ and $XCO_2$ observations (see section 2.1) is shown in Figures 5 to 7.

    At the surface there is an overall substantial reduction of RMSE between 80 km and 9 km (i.e. between 1.8 ppm and 3.5 ppm for hourly data) which is clearly not linear (Figs 5a and 5b). The RMSE difference between the 80 km and 40 km simulations or the 40 km and 25 km simulations is not as large as the difference between the higher resolution simulations (e.g. 25 km and the 16 km or the 16 km and the 9 km). This is particularly pronounced for the daily maximum $CO_2$ occuring usually at
nighttime, which is generally controlled by local fluxes and small-scale transport of tracers and therefore it is more sensitive to resolution. The daily maximum values are generally much better captured at 9 km resolution compared to 80 km with a reduction in the RMSE of around 2.5 ppm in January and 6 ppm in July. Indeed, there are large differences between the RMSE of the daily maximum and minimum $CO_2$ values. As expected, daily minimum values that emerge during daytime have smaller RMSE. This is because during daytime the minimum $CO_2$ values are influenced by the larger-scale fluxes and tracer transport which are less sensitive to high resolution. The reduction in RMSE of the daily minimum $CO_2$ is therefore smaller than for
the daily maximum, but it is still considerable with an RMSE decrease of around 0.75 ppm from 80 km to 9 km resolutions in both January and July. These differences reflect on the ability of the model to represent the diurnal cycle. The 9 km simulation clearly shows a general improvement in the $CO_2$ diurnal cycle near the surface, with smaller differences in the RMSE of the




two daily extremes. The largest RMSE reduction comes from mountain sites (over 1000 m above mean sea level), ranging between 6 and 10 ppm for hourly $CO_2$ (Figs 5c and 5d) compared to the lowland sites which can see improvements between 0.5 to 2 ppm for hourly $CO_2$ RMSE near the surface (Figs 6c and 6d).

In general there is also a notable reduction in the spread of the RMSE at the different sites with resolution, as shown by the $\sigma$ RMSE values below the panels in Fig 5 and 6. This implies that the spatial gradients between stations are better represented at higher resolutions. The global mean correlation coefficient also increases with resolution from 0.47 to 0.56 in January and 0.51 to 0.59 in July for the hourly $CO_2$, with consistently higher correlations for the daily mean, minimum and maximum $CO_2$ at higher resolution.

As expected, the sensitivity to the strategy of sampling the model level at observing stations is generally small over lowlands but large over mountains, particularly at low resolution (Fig. S1). At mountain sites, the model level at the real station height above mean sea level is predominantly in the free troposphere and therefore it has a small sensitivity to the local fluxes/flow; whereas taking the model level with respect to the model ground generally exhibits larger errors associated with local influences in the boundary layer. The difference between the two sampling strategies on the RMSE and correlation coefficients becomes smaller at high resolution (Fig. S2). This reflects an improvement in the capability of the model to represent the flow and fluxes around complex topography at higher horizontal resolution.

The $XCO_2$ RMSE at the TCCON sites also displays a general decrease with resolution (Fig. 7), with differences of the order of 0.1 ppm from 80 km to 9 km resolutions and increases in the correlation coefficients of up to 0.05.

It is likely that the larger footprint of $XCO_2$ (Belikov et al., 2017) at most TCCON stations – associated with its sensitivity to large-scale flux patterns – (Keppel-Aleks et al., 2011, 2012) is causing most TCCON sites to be less sensitive to horizontal resolution. However, there is a large variation in RMSE between sites (see $\sigma$ RMSE in Fig. 7 and Fig S6) which is reduced at high resolution. In particular, the TCCON site at Pasadena (California, USA) located near the anthropogenic emission hotspot of the Los Angeles megacity stands out (Fig. S6). The improvement associated with high resolution at Pasadena is indeed remarkable in January (i.e. approximately 2 ppm RMSE reduction). A more detailed study for Pasadena is provided in section 3.5.

The change of RMSE with resolution is partly associated with the improvement in the transport and also the representation of the local fluxes at higher resolutions. Figure 8 shows that when the fluxes are switched off during the 1-day forecast, there is still an improvement with resolution at most sites, but the magnitude of the error reduction is smaller (see symbols to the right of the dash line). This is very clear for a large number of mountain sites and TCCON sites affected by anthropogenic emissions such as Pasadena (USA) in January and Saga (Japan) in July. However, there are also some sites and months in which the impact of resolution is better without fluxes than with fluxes (e.g. Pasadena in July). This would indicate that in this case the errors in the fluxes are the main cause of the deterioration in RMSE with resolution.

The overall global error statistics of the 9 km and 80 km simulations including the systematic (or bias) error and the standard (or random) error are shown in Table 3. The reduction in RMSE at 9 km is associated with a decrease in the magnitude of the $CO_2$ biases on average of 1.5 to 2 ppm near the surface and up to 0.2 ppm for $XCO_2$ and a general reduction in the $CO_2$ random error of 1 to 1.5pm near the surface and 0.1 ppm for $XCO_2$ (Figs S3 and S4). The biases depend largely on the bias



of the $CO_2$ initial conditions, as well as the biases of the fluxes and tracer transport. What is important in this sensitivity study is that the standard deviation of the bias at each station – i.e. the inter-station bias – is reduced at 9 km with respect to 80 km, as shown by the shaded area in Figs S3 and S4. The largest decrease in the inter-station bias between 80 km 9 km occurs in January, when it is almost halved near the surface. The errors at the individual observing stations are listed in Tables S1, S2,
S3, S4.

### 3.3    Impact of horizontal resolution on $CO_2$ forecast error growth

In 10 days the global mean RMSE of $CO_2$ forecast at the in situ surface stations grows by around 1.4 ppm in January and around 1 ppm in July (Fig. 9). It is worth noting that this error growth is smaller in magnitude than the impact of increasing horizontal resolution from 80 km to 9 km. Namely, the 10-day forecast at 9 km is better than the 1-day forecast at 80 km near
the surface. At the TCCON sites the $XCO_2$ RMSE grows on average between 0.2 and 0.5 ppm in 10 days (Fig. 10). The forecast RMSE growth for near-surface $CO_2$ and $XCO_2$ does not appear to be linear, with a slow growth until day-4, and faster increase from day 5 onwards. The $CO_2$ RMSE growth at 80 km is slightly faster than at 9 km. In summary, the gain in skill from horizontal resolution is maintained throughout the 10-day forecast. Thus, the results suggests that the horizontal resolution has a small but positive impact on the short and medium-range forecast skill for $CO_2$ and $XCO_2$.

As expected, the RMSE in July is largest because of the high uncertainty associated with the modelled biogenic fluxes at synoptic scales which influence the variability at continental sites (Agustí-Panareda et al., 2014). There is also a larger uncertainty in the meteorology driving the tracer transport during summer compared to winter (Haiden et al., 2017). The fact that the forecast RMSE at day-1 is larger than at day-2 in July is associated with a sporadic overestimation of daily maximum $CO_2$ peaks at sites influenced by strong local fluxes. There are several potential causes of the overestimation (e.g. biogenic
fluxes responding to rapid adjustments in meteorology after analysis re-initialisation at 00 UTC or issues with the tracer transport associated with short spin up period), but these are beyond the scope of this study.

The near-surface $CO_2$ RMSE increase during the forecast appears to come mostly from an increase in random error in January and from both mean and random error in July (Fig. S3); whereas for $XCO_2$, both mean and random errors contribute equally to the forecast RMSE growth in January and July (Fig. S4). This is probably linked to the distribution of the stations,
as most in situ stations are located in the northern hemisphere; whereas TCCON stations are more equally distributed in both hemispheres, and thus, the mean error at all stations does not show differences between summer and winter conditions.

### 3.4    Impact of horizontal resolution on $CO_2$ small-scale variability

The sensitivity of the RMSE to resolution is generally associated with regions that are affected by small-scale variability that cannot be properly represented by typical global tracer transport models (Law et al., 2008; Patra et. al, 2008). Figure 11 shows
that the mean small-scale variability, given by the standard deviation within 1 degree x 1degree grid box, can be as large as 10 ppm near emission hotspots at the surface during daytime. Larger values than 10 ppm can be found over most land areas at nighttime (Fig 12). These values are likely to be underestimated, since we expect horizontal gradients to become steeper as the





resolution increases, the point sources associated with anthropogenic activities become stronger at the grid cell scale and part of the sub-grid scale flow is resolved.

Coastal sites and mountain sites have a typical sub-grid scale variability of around 5 ppm during daytime. This variability varies from January to July, depending on meteorological conditions (e.g. stagnant or windy conditions) and the magnitude/sign of fluxes (e.g. biogenic activity shifting northwards in northern hemisphere summer). Over land, the patterns of sub-grid scale variability of surface and total column are consistent (Figs 11 and 13), as both are subject to surface heterogeneity in terms of topography and fluxes. However, there is a difference in magnitude because the variability of the total column average is much smaller than the variability at the surface.

$XCO_2$ has a maximum standard deviation of 1 ppm near surface flux hotspots and typically less than 0.5 ppm in most regions (Fig. 13), which is consistent with other estimates from regional studies (Corbin et al., 2008; Pillai et al., 2010). The differences in the small-scale $XCO_2$ variability between day and night appear to be small. Interestingly, the small-scale variability of $XCO_2$ is much larger in summer than in winter (both in northern and southern hemispheres). During the growing season, negative $CO_2$ anomalies associated with plant photosynthesis and positive anomalies associated with ecosystem respiration and anthropogenic emissions combine to create steeper gradients throughout the troposphere – as illustrated in Fig. 1b – that contribute to the enhanced sub-grid scale variability in summer compared to winter. Over the ocean, the small-scale variability of $XCO_2$ ranges between 0.1 and 0.3 ppm, with lower values in the winter and higher values in the summer. In the northern hemisphere summer, the values over the ocean and over the land are comparable. Whereas near the surface, the mean sub-grid scale variability is an order of magnitude smaller over the ocean than over land. This is because over land the surface fluxes dominate the gradients resulting in the steepest gradients being near the surface; while over the ocean, the transport associated with the weather systems creates steep $CO_2$ gradients in the free troposphere. Therefore, column averaged $CO_2$ is much more likely to be influenced by sub-grid scale variability associated with weather systems than by surface $CO_2$ fluxes over the ocean.

## 3.5 Example of horizontal resolution impact at an urban site

Although the winds, the topography and the spatial heterogeneity of the fluxes are generally better represented at high horizontal resolution, there can still be a deterioration in the RMSE scores at sites where the local influence is strong and the emissions/biogenic fluxes have large errors in the model. In this section we present an example of such a case at the Caltech TCCON site in Pasadena (California, USA, see Tab A2) with $XCO_2$ under clear-sky and daylight conditions. The variability of the simulated $XCO_2$ exhibits a substantial improvement with high resolution in winter and an equally considerable deterioration in summer (Fig. 15). Thus, it illustrates some of the challenges associated with urban regions.

Pasadena is located 14 km north-east of the megacity of Los Angeles (LA) with a large local anthropogenic emission influence (Wunch et al., 2009; Newman et al., 2016). The $XCO_2$ variability in the model is also mainly explained by the local anthropogenic emissions (Figs S7 and S8) producing very large $CO_2$ enhancements in the Planetary Boundary Layer (PBL) (Fig. S9) and therefore in $XCO_2$. The $CO_2$ budget of the anthropogenic emissions used at 9 km and 80 km is the same. However, the instantaneous values of the emissions per square meter are much higher at 9 km than at 80 km, representing some of the steep gradients and heterogeneous distribution of fossil fuel emissions within the LA basin, with higher emissions in



downtown LA and lower emissions in Pasadena (e.g. Feng et al., 2016). At 80 km, Pasadena and downtown LA are in the same model grid box, which means this gradient cannot be represented. In addition to the influence of anthropogenic emissions, the seasonal variation of the winds is very pronounced in Pasadena, with a large contrast in the origin or air masses between winter and summer (Verhulst et al., 2017).

In winter, air masses originate from various directions: from the prevailing westerly and southerly winds bringing and accumulating polluted air from the LA megacity, to northerly and easterly flow characterised by cleaner air with lower $CO_2$ values from the surrounding desert and mountains (Newman et al., 2016). Persistent low wind conditions lead to a large accumulation of $CO_2$ in the LA basin as it remains trapped by the mountains. These episodes results in large enhancements in $XCO_2$ (Hedelius et al., 2017) coinciding with the high $CO_2$ anomalies over periods of a few days (e.g. 26 to 30 January in Fig. 15a). In those stagnant conditions, the 9 km simulation is in much closer agreement with the observed $XCO_2$ peaks than 80 km simulation, which overestimates the $XCO_2$ anomalies. This is because at 80 km resolution there is an effectively uniform emission for the whole LA basin. Note that the $CO_2$ and $XCO_2$ small-scale variability around LA appears to be larger in winter than in summer (Figs 11 and 14). Without preserving the sharp gradient in emissions between Pasadena an downtown LA, the $CO_2$ accumulation is overestimated in Pasadena.

The atmospheric circulation in summer is mainly controlled by the sea-mountain breeze (Lu and Turco, 1994). Daytime advection of anthropogenic $CO_2$-rich air from LA city results in $XCO_2$ peaking in the afternoon before it is vented over the mountains (Newman et al., 2013, 2016). The overestimation in the summer $XCO_2$ peaks at 9 km is likely reflecting an overestimation of the emissions in downtown LA. The enhancement of $CO_2$ from anthropogenic emissions is larger at 9 km than at 80 km (Fig. S8). This suggests an overestimation of the hotspot emissions over the LA basin in the temporally-extrapolated EDGAR inventory, which is smoothed and less noticeable at lower resolution. There are many reasons why the anthropogenic emissions used in the model can be overestimated, including the temporal extrapolation based on country-scale scaling factors and the use of annual constant emissions in EDGAR4.2FT2010 instead of seasonally varying emissions (Corbin et al., 2010).

Differences in the sampling location (centre of grid is 3 km and 34 km from station location at 9 km and 80 km respectively), orography (15 m below and 46 m below the station height at 9 km and 80 km respectively) as well as differences in flow and local biogenic fluxes can also play a role in explaining the differences between the simulations at 80 km and 9 km resolutions. The results are consistent with previous studies by Feng et al. (2016) and Hedelius et al. (2017). They found that uncertainties in the fluxes and their high resolution representation in the LA basin are as important as the atmospheric tracer transport in the representation of the $CO_2$ enhancement and its variability in Pasadena.

This example at Pasadena highlights the importance of horizontal resolution to represent local gradients of $CO_2$ fluxes in order to reduce the atmospheric $CO_2$ representativeness error. It emphasizes that the impact of increasing horizontal resolution is not only to reduce the error of atmospheric $CO_2$ simulations, but to enhance the sensitivity of the modelled atmospheric $CO_2$ variability to the $CO_2$ fluxes in urban regions characterised by emission hotspots. Therefore horziontal resolution is crucial for atmospheric inversion systems that aim to estimate anthropogenic emissions.





## 4    Discussion and conclusions

This paper addresses the importance of horizontal resolution in the representation of $CO_2$ variability at diurnal and synoptic scales, referred to here as $CO_2$ weather. The $CO_2$ simulations performed with the ECMWF IFS model allow to quantify the combined impact of horizontal resolution associated with (i) the on-line modelling of the winds, (ii) the numerical tracer transport model and (iii) the spatial/temporal distribution of $CO_2$ fluxes over land. The assessment is done by comparing the model errors at various horizontal resolutions with respect to a wide range of observations with hourly resolution and distributed around the globe. The horizontal resolution of the model ranges from 9 km – as in current operational high resolution weather and $CO_2$ forecasts at ECMWF – and 80 km, which corresponds to the ERA-Interim re-analysis resolution, widely used by many off-line tracer transport models. The conclusions to the three main questions addressed in the paper are summarised below:

1. **What is the sensitivity of the modelled atmospheric $CO_2$ variability at diurnal and synoptic timescales to horizonal resolution?**

   The high horizontal resolution of 9 km leads to a general improvement in the simulated variability of hourly near-surface and column-averaged atmospheric $CO_2$ compared to the resolution of 80 km. This is shown by a reduction in the mean RMSE of around 1.8 ppm in winter and 3.5 ppm in summer (equivalent to 33% error reduction) and 0.1 ppm (i.e. around 10% error reduction) at in situ and TCCON sites respectively, which is associated with a reduction of both the mean and random errors in the model. The inter-station variability is also generally improved in the 9 km simulation for near-surface and column-averaged $CO_2$ in January and July, with the standard deviation of station biases reduced up to 50% compared to the 80 km simulation in January for near surface $CO_2$.

   Column-averaged $CO_2$ is not as sensitive to horizontal resolution as near-surface $CO_2$ because it has a larger footprint or area of flux influence, except for sites like Pasadena which are close to $CO_2$ emission hotspots. Similarly, minimum daily values of atmospheric $CO_2$ are less sensitive to the horizontal resolution than maximum daily values because their footprint tends to be larger in size.

   This study also shows that the RMSE reduction of error with horizontal resolution is not linear. This implies that results from sensitivity studies exploring the impact of resolution based on coarse simulations which show small sensitivity to horizontal resolution cannot be extrapolated to higher horizontal resolutions. These results are consistent with the findings of the Lin et al. (2017) study based on wider range of model resolutions from $\sim$ 100 km down to 1 km and observations at three mountain sites. The reduction in model error associated with the increase of horizontal resolution to 9 km emanates from four different well-known and connected aspects, namely:

   (a) Better accuracy of the horizontal winds. The strength of the winds determines the observed $CO_2$ variability – i.e. the detected $CO_2$ enhancement – close to emission hotspots like in urban regions (Newman et al., 2013; Xueref-Remy et al., 2018). Therefore, the error in the wind will affect the value of the enhanced $CO_2$ as much as the error in the fluxes. In this context, for example, a wind speed error reduction of 0.5 ms$^{-1}$ – as shown in sec. 3.1 – across





a gradient of $10\,\mathrm{ppm}/1$ degree – typical of urban areas as shown in section 3.4 – throughout a 6-hour period can result in a $CO_2$ error reduction of around $1\,\mathrm{ppm}$. Uncertainty in the winds has been shown to be one of the largest contributors to the uncertainty in the estimated fluxes over urban areas (e.g. Hedelius et al., 2017).

(b) An overall reduction of the numerical error associated with lower spatial and temporal truncation errors, leading to a reduction in tracer advection errors (Prather et al., 2008).

(c) A general improvement in the horizontal and vertical sampling at the station locations in the model associated with a more realistic representation of orography and coastal boundaries.

(d) More realistic representation of $CO_2$ flux distribution at the surface. High resolution gives an increased capability to represent small-scale sharp gradients associated with complex topographical boundaries at coastal and mountainous terrain sites, as well as the presence of strong local surface fluxes of $CO_2$ such as anthropogenic emission hotspots.

2. **How is the horizontal resolution affecting the forecast error growth of atmospheric $CO_2$?**

The horizontal resolution has a consistent positive impact on the error reduction at all forecast lead times, from day 1 to day 10, implying a long-lived improvement in the prediction skill. The RMSE growth is small up from days 1 to 4, namely less than $0.5\,\mathrm{ppm}$ near the surface $CO_2$ and less than $0.05\,\mathrm{ppm}$ for $XCO_2$. Over the 10 days there is an increase in RMSE of 1 to $1.5\,\mathrm{ppm}$ at the surface and 0.1 to $0.5\,\mathrm{ppm}$ for the total column. This error growth is not linear. For example, in July the error of the 1-day forecast is worse than the 2-day forecast, with a slower error increase during the 2 to 4 day forecast and a generally faster error increase from day 5 to day 10 in the forecast. This incoherent change in the error evolution at the beginning of the forecast is likely linked to the strong influence of the biogenic surface fluxes, which respond very fast to changes in temperature, moisture and radiation forcing in the model. Inconsistencies between the analysis as initial condition and the model forecast can cause spin up adjustments which may lead to a degradation of the 1-day forecast.

Generally, the improvement of forecast skill with increased horizontal resolution is most pronounced in January, when at $9\,\mathrm{km}$ resolution the skill of the 10-day forecast is better or equal to the accuracy of the 1-day forecast at $80\,\mathrm{km}$ both near the surface and for the column average $CO_2$. It is likely that the skill of the 10-day forecast to represent variability of $CO_2$ during summer conditions is hampered by the growing errors in the surface biogenic fluxes during the forecast, as they can be an important contributor to synoptic variability in the summer (Agustí-Panareda et al., 2014).

3. **Where and when are the typical representativeness errors associated with unresolved small-scale variability largest?**

During daytime, the $CO_2$ small-scale variability of the $9\,\mathrm{km}$-resolution forecast ranges from $1\,\mathrm{ppm}$ to $10\,\mathrm{ppm}$ at the surface and an order of magnitude smaller ($0.1\,\mathrm{ppm}$ to $1\,\mathrm{ppm}$) for the total column average. It points to the areas associated with small-scale gradients where horizontal resolution matters: coastal boundaries and mountain regions have typical values of 5ppm/degree and $CO_2$ flux hotspots have the highest variability of up to $10\,\mathrm{ppm}/$ degree. During





nighttime, the small-scale variability tends to be larger than $10\,\mathrm{ppm}$ over most areas over near the surface; whereas that of column-average $CO_2$ shows small differences between day and night.

The high horizontal resolution gives us an insight on the areas with high sensitivity to uncertainty associated with both local tracer transport and fluxes. It is in these areas where improvements in the tracer transport and increased
understanding of the heterogeneity and complexity of the surface will be crucial in the future model developments. Since these areas are close to emission hotspots, it is clear that in order to monitor $CO_2$ emissions, particularly from cities and power stations such as in the new Carbon Human Emission project (www.che-project.eu), it is paramount to invest in high horizontal resolution models.

Interesting differences are found between surface and column-averaged variability. Near the surface the variability is
most pronounced close to emission hotspots and complex terrain. For column-averaged $CO_2$ the sub-grid scale variability is also substantial over the ocean downstream from emissions. This emphasizes the importance of the tranport influence on $XCO_2$ variability. Small-scale variability is also found to be more pronounced in summer than in winter, as biogenic $CO_2$ fluxes of opposite sign in summer enhance the $CO_2$ gradients in the atmosphere.

In summary, this paper has shown that model simulations using the CAMS $CO_2$ forecasting system at $9\,\mathrm{km}$ resolution can
provide a more accurate representation of tracer transport and the local influences of surface fluxes than at lower resolutions ranging from $80\,\mathrm{km}$ to $16\,\mathrm{km}$, resulting in an overall better representation of the atmospheric $CO_2$ variability at diurnal and synoptic time scales. However, at higher horizontal resolution there is also higher sensitivity of atmospheric $CO_2$ to $CO_2$ flux errors, as emissions and biogenic flux hotspots are not diffused over large areas like in lower resolution models. Thus, higher resolution models also risk the deterioration in the forecast RMSE, e.g. near emission hotspots associated with larger errors.
With the enhancement of the model uncertainty at high resolution, the prospect of further increasing the horizontal resolution needs to be carefully balanced with improvements in the most uncertain model processes.

The impact of horizontal resolution on the accuracy of the winds highlights that errors in the wind need to be considered as an important source of uncertainty both in the atmospheric $CO_2$ analysis/forecast as well as in the inversion systems (Polavarapu et al., 2016). The findings in this study also suggest that increasing horizontal resolution up to kilometric scales in atmospheric
data assimilation and inversion systems would allow the use of more in situ and high resolution satellite observations close to strong sources/sinks and over complex terrain. Lin et al. (2017) found that a horizontal resolution of at least $4\,\mathrm{km}$ is required to simulate a realistic diurnal cycle of $CO_2$ at mountain sites.

Currently, the precision of $XCO_2$ from satellite observations is around 1.0 to $1.5\,\mathrm{ppm}$ for ACOS-GOSAT data (O'Dell et al., 2012) and OCO-2 data (Wunch et al., 2017). However, if tracer transport models cannot represent their variability accurately
in space and time, all the efforts to reduce the errors from the satellite retrievals of $CO_2$ will not be fruitful in their attempt to reduce the uncertainty in the estimation of surface fluxes. This is because relatively small differences in atmospheric mixing ratios are associated with significant differences in surface fluxes (Houweling et al., 2010; Ott et al., 2015). The benefits of high resolution in inversion systems will also need to be balanced with the costs of running a model at such high resolution.





Finally, the CAMS high resolution forecast running currently at 9 km resolution can provide benchmarks for other simulations using coarser grids or off-line meteorology (Yu et al., 2018). Both CAMS analysis and high resolution forecasts are freely available to users (https://atmosphere.copernicus.eu). Potential applications include the estimation of representativeness errors and data selection screening of observations from satellites and in situ stations in data assimilation systems, spatial collocation

5   of $XCO_2$ from satellite and TCCON data for validation purposes (e.g. Guerlet et al., 2013) or as boundary conditions for high resolution simulations and/or inversions at regional scales.

*Data availability.* The data is accessible by contacting the corresponding author (Anna.Agusti-Panareda@ecmwf.int).



**Table 1.** Datasets and models of $CO_2$ fluxes used in simulations listed in Tab. 2.

| Flux type | Source | Temporal resolution | Resolution (lat x lon) [degrees$^2$] | Reference |
|---|---|---|---|---|
| Anthropogenic | EDGARv4.2FT2010 | Annual mean | 0.1x0.1 | Olivier and Janssens-Maenhout (2015) |
| Biomass burning | GFAS | Daily mean | 0.1x0.1 | Kaiser et al. (2012) |
| Ocean | Takahashi climatology | Monthly mean | 4.0x5.0 | Takahashi et al. (2009) |
| NEE | CHTESSEL | Adapted to model time step | Adapted to model resolution | Boussetta et al. (2013), Agustí-Panareda et al. (2016) |

**Table 2.** List of simulations with different resolutions given by different model grids. All simulations use 137 vertical model levels. All the experiments have been performed in January and July 2014.

| Experiment | Model resolution | Model grid | Model time step | Surface $CO_2$ fluxes |
|---|---|---|---|---|
| 9km-EXP | 9 km | Tco1279 | 7.5 min | Table 1 |
| 16km-EXP | 16 km | Tco639 | 12 min | Table 1 |
| 25km-EXP | 25 km | Tco399 | 15 min | Table 1 |
| 40km-EXP | 40 km | Tco255 | 20 min | Table 1 |
| 80km-EXP | 80 km | Tl255 | 45 min | Table 1 |



**Table 3.** Surface $CO_2$ and $XCO_2$ mean statistics for bias, standard error and RMSE of all stations and the standard deviation of inter-station statistics (in brackets and bold phase) from the highest and lowest resolution simulations, i.e. 9km-EXP and 80km-EXP respectively. All the SFC $CO_2$ stations used in January and July are listed in Tabs S1 and S2; the TCCON stations used in the $XCO_2$ statistics are listed in Tabs S3 and S4.

| Data | Period | Temporal Resolution | Bias | | STDerror | | RMSE | |
|---|---|---|---|---|---|---|---|---|
| | | | 9km-EXP | 80km-EXP | 9km-EXP | 80km-EXP | 9km-EXP | 80km-EXP |
| SFC $CO_2$ | January | Hourly mean | -0.09 (**2.23**) | 1.73 (**4.65**) | 2.70 (**2.23**) | 3.78 (**3.25**) | 3.21 (**2.63**) | 4.96 (**4.99**) |
| | | Daily mean | -0.06 (**2.32**) | 1.76 (**4.66**) | 1.76 (**1.42**) | 2.59 (**2.18**) | 2.43 (**2.15**) | 4.08 (**4.44**) |
| | | Daily min | -0.30 (**1.47**) | 0.62 (**2.41**) | 1.41 (**1.08**) | 1.82 (**1.37**) | 1.87 (**1.38**) | 2.59 (**2.16**) |
| | | Daily max | 0.02 (**4.39**) | 2.66 (**7.74**) | 3.68 (**3.96**) | 4.73 (**4.52**) | 4.77 (**5.07**) | 7.12 (**7.68**) |
| SFC $CO_2$ | July | Hourly mean | 0.96 (**5.70**) | 2.67 (**9.54**) | 6.70 (**6.87**) | 9.64 (**11.82**) | 7.90 (**7.94**) | 11.56 (**14.04**) |
| | | Daily mean | 1.14 (**6.48**) | 2.84 (**9.70**) | 4.08 (**5.40**) | 5.52 (**8.22**) | 5.91 (**7.36**) | 8.55 (**11.27**) |
| | | Daly min | 1.48 (**4.12**) | 0.86 (**4.76**) | 3.70 (**4.35**) | 4.11 (**3.70**) | 4.87 (**5.31**) | 5.59 (**4.78**) |
| | | Daily max | -0.44 (**13.95**) | 5.07 (**22.82**) | 9.10 (**11.79**) | 12.20 (**19.38**) | 13.22 (**15.55**) | 19.63 (**26.18**) |
| $XCO_2$ | January | Hourly mean | -0.49 (**0.72**) | -0.38 (**0.90**) | 0.58 (**0.26**) | 0.69 (**0.52**) | 1.02 (**0.36**) | 1.12 (**0.66**) |
| | | Daily mean | -0.51 (**0.75**) | -0.38 (**0.96**) | 0.47 (**0.22**) | 0.58 (**0.46**) | 0.97 (**0.39**) | 1.09 (**0.66**) |
| | | Daily min | 0.60 (**1.06**) | 0.75 (**1.15**) | 0.92 (**0.50**) | 1.03 (**0.58**) | 1.36 (**0.85**) | 1.53 (**0.98**) |
| | | Daily max | -1.60 (**0.98**) | -1.49 (**1.19**) | 1.00 (**0.75**) | 1.13 (**0.89**) | 1.99 (**1.07**) | 2.11 (**1.11**) |
| $XCO_2$ | July | Hourly mean | 1.01 (**0.58**) | 1.04 (**0.69**) | 0.71 (**0.31**) | 0.74 (**0.28**) | 1.28 (**0.56**) | 1.35 (**0.61**) |
| | | Daily mean | 0.99 (**0.57**) | 1.03 (**0.65**) | 0.56 (**0.27**) | 0.59 (**0.26**) | 1.18 (**0.54**) | 1.25 (**0.59**) |
| | | Daily min | 2.09 (**0.81**) | 2.18 (**0.89**) | 1.05 (**0.65**) | 1.07 (**0.67**) | 2.37 (**0.97**) | 2.46 (**1.03**) |
| | | Daily max | -0.01 (**0.84**) | -0.07 (**0.75**) | 0.92 (**0.47**) | 0.87 (**0.43**) | 1.21 (**0.54**) | 1.15 (**0.44**) |





**Figure 1.** Snapshots of averaged-column $CO_2$ ($XCO_2$) [ppm] above (in reds) and below (in greens) the global mean on 15 January (upper panel) and 15 July (lower panel) at 12 UTC from the Copernicus Atmosphere Monitoring Service global $CO_2$ forecast.





**Figure 2.** Schematic of $CO_2$ simulations with cyclic forecast configuration with 10-day forecasts initialised every day from 1st of the month to the 10th day of the following month. Initial conditions are depicted by arrows (see legend) and period of evaluation in which several forecast lead times can be compared is delimited by the red dashed line. The standard $CO_2$ simulations are composed by a series of 1-day forecasts as shown by the green rectangles.



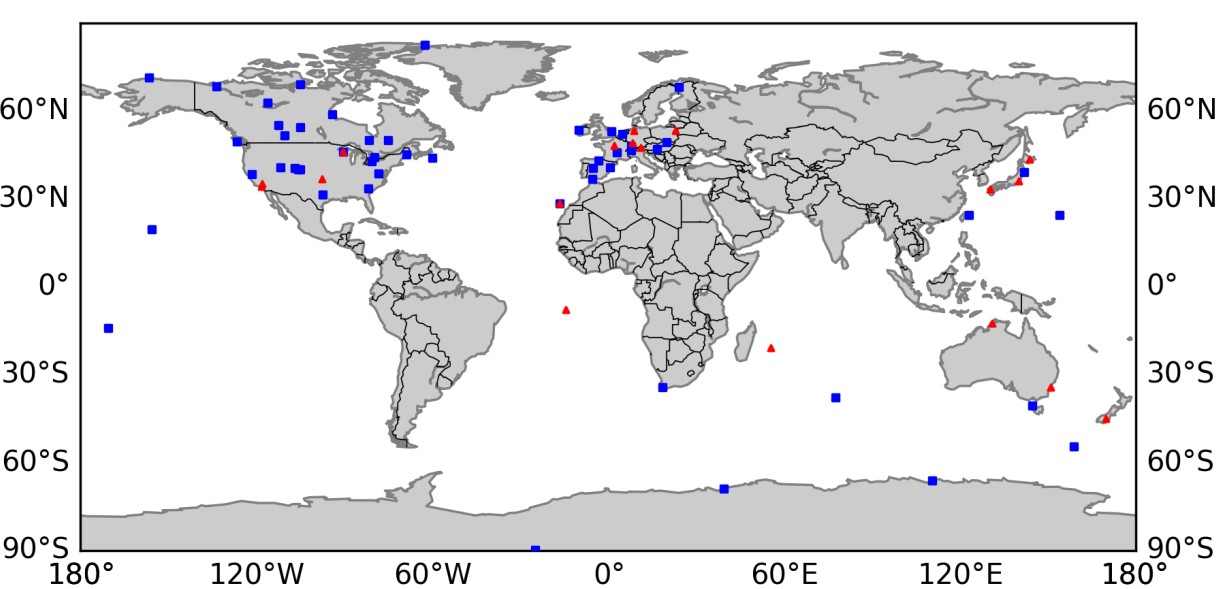

**Figure 3.** Map of in situ (blue squares) and TCCON (red triangles) stations. Detailed information on each station is provided in Tabs. A1 and A2.





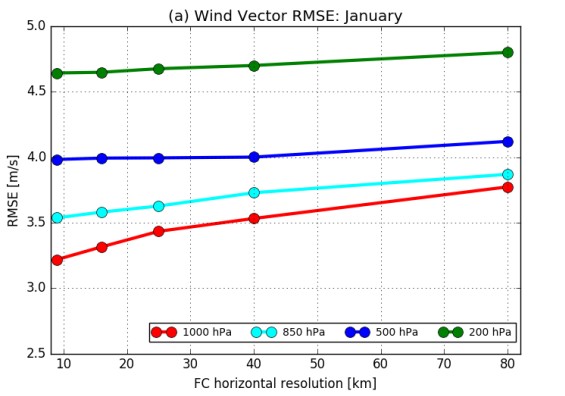
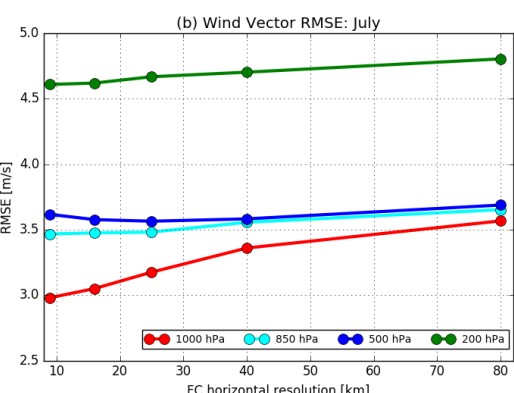

**Figure 4.** Mean RMSE of vector wind [m/s] at different model resolutions in (a) January and in (b) July for around 400 radiosonde stations over the globe. Different cololours represent different pressure levels (see legend). All the model simulations are based on the standard 1-day forecast configuration shown in Fig. 2.





**Figure 5.** Mean RMSE of near-surface $CO_2$ [ppm] (a,b) and pearson's correlation coefficient R (c,d) at different model resolutions in January (left panels) and July (right panels) for all 51 stations (see Tab. A1). The standard deviation of the RMSE from each station is shown by the numbers below the horizontal resolution for each temporal resolution (hourly, daily mean, daily min and daily max). All the model simulations are based on the 1-day forecast. Note that different scales are used in each panel.



**Figure 6.** Mean RMSE of near-surface $CO_2$ [ppm] at different model resolutions in January (left panels) and July (right panels) for (a,b) 37 lowland stations (below 1000 m above sea level) (c,d) 12 mountain stations (1000 m above sea level and higher, excluding bao and spo, as listed in Tab. A1). The standard deviation of the RMSE from each station is shown by the numbers below the horizontal resolution for each temporal resolution (hourly, daily mean, daily min and daily max). All the model simulations are based on the 1-day forecast. Note that different scales are used in each panel.





**Figure 7.** Mean RMSE of $XCO_2$ [ppm] and Pearson's correlation coefficient R with respect to observations from 18 TCCON stations (see Tab. A2) at different model resolutions in (a) January and (b) July. The standard deviation of the RMSE from each station is shown by the numbers below the horizontal resolution for each temporal resolution (hourly, daily mean, daily min and daily max). All the model simulations are based on the 1-day forecast. Note that different scales are used in each panel.





**Figure 8.** Sensitivity of horizontal resolution impact to surface $CO_2$ fluxes at (a,b) surface stations and (c,d) TCCON stations in January and July, as shown by the difference in RMSE between different $CO_2$ tracer, i.e. the standard tracer ($RMSE_{9km} - RMSE_{80km}$) and the $CO_2$ tracer with just transport ($RMSE_{NFX9km} - RMSE_{NFX80km}$) in y-axis and x-axis respectively. The symbols that are close to the dash line correspond to stations that have a small sensitivity to local fluxes; while at the stations associated with symbols that are located above/below the dash line there is a negative/positive contribution of the local fluxes at high resolution. The further from the dash line, the larger the contribution of the local fluxes. The stations located on top of the y-axis are mainly impacted by local fluxes. The surface stations in (a,b) are depicted with different symbols depending on whether they are classified as mountain, continental, coastal or remote (see Table A1), while TCCON stations that are strongly influenced by fluxes are labelled with station name. Note that different scales are used in each panel.




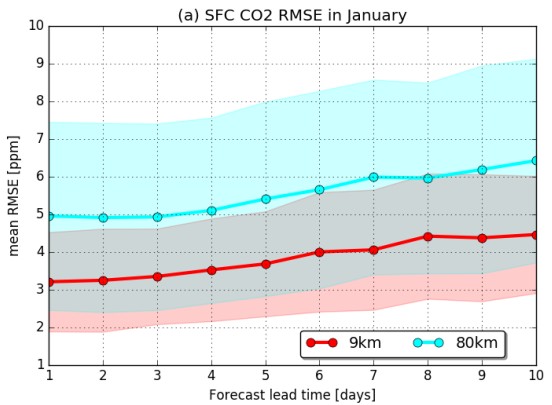
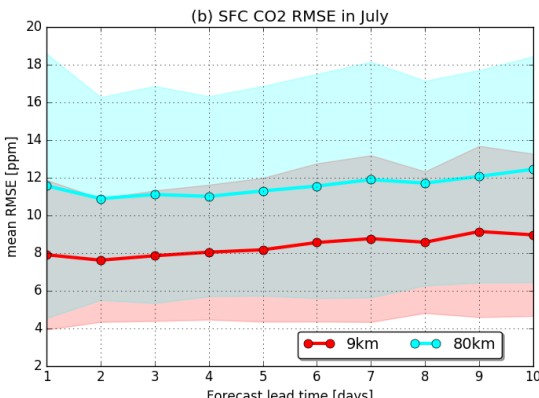

**Figure 9.** Mean RMSE of near-surface $CO_2$ [ppm] at different forecast lead times for the 9km-EXP (red) and 80km-EXP (blue) in (a) January and (b) July. The errors are computed with respect to hourly continuous in situ surface measurements from 51 stations (see Tab A1). The error standard deviation between the different stations is shown with the shaded area: red for $9\,km$, blue for $80\,km$ and grey for overlap. Note that different scales are used in each panel.

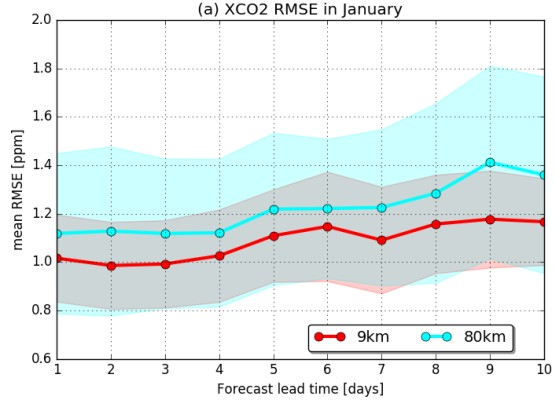
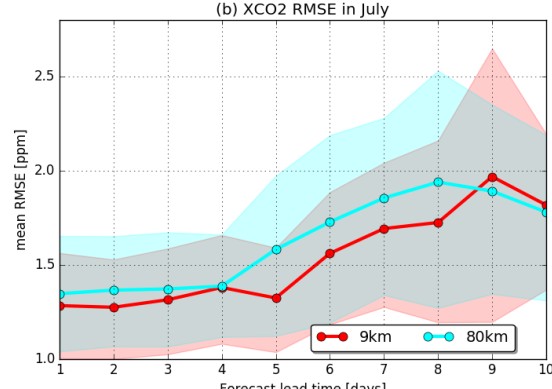

**Figure 10.** Same as 9 but for $XCO_2$ [ppm]. The errors are computed with respect to hourly TCCON observations from 18 TCCON stations (see Tab. A2).





**Figure 11.** Monthly mean surface $CO_2$ small-scale variability ($\sigma$) within 1 degree x 1 degree grid boxes [ppm] at 13:00 local time in (a) January and (b) July.





**Figure 12.** Monthly mean surface $CO_2$ small-scale variability ($\sigma$) within 1 degree x 1 degree grid boxes [ppm] at 01:00 local time in (a) January and (b) July.





**Figure 13.** Monthly mean $XCO_2$ small-scale variability ($\sigma$) within 1 degree x 1 degree grid boxes [ppm] at 13:00 local time in (a) January and (b) July.





**Figure 14.** Monthly mean $XCO_2$ small-scale variability ($\sigma$) within 1 degree x 1 degree grid boxes [ppm] at 1:00 local time in (a) January and (b) July.




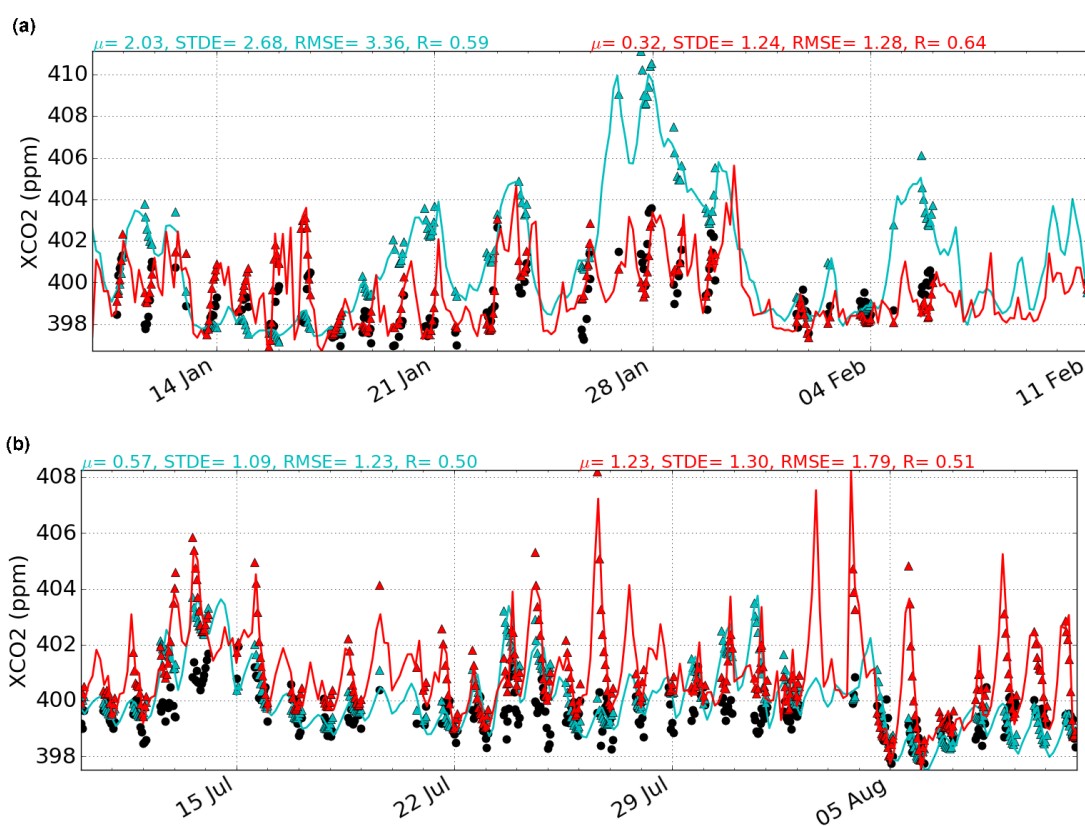

**Figure 15.** Hourly $XCO_2$ [ppm] at TCCON site of Pasadena (CA, USA) in (a) January and (b) July from 80km-EXP (blue) and 9km-EXP (red) andsimulations. Hourly observations are shown black circles. Triangles represent the model XCO2 after smoothing with TCCON averaging kernel and prior. The bias ($\mu$), standard error ($STDE$) and Pearson correlation coefficient ($R$) from 80km-EXP (blue) and 9km-EXP (red) are shown at the top of each panel.

Table A1: Continuous in situ stations (surface and tower) used to evaluate synoptic variability. N/A denotes reference not available. The full names for the abbreviations of the network organizations are provided in Tab. A3.

| Station ID | Lat/Lon [degree] | Altitude [masl] | Intake height [magl] | Network | Reference | Type |
|---|---|---|---|---|---|---|
| brw | 71.32N/-156.61 | 11 | 16 | NOAA | Peterson et al. (1986) | coastal |
| mlo | 19.54N/-155.58 | 3397 | 40 | NOAA | Thoning et al. (1989) | mountain |
| smo | 14.25S/-170.56 | 42 | 10 | NOAA | Halter et al. (1988) | remote |
| spo | 89.98S/-24.80 | 2810 | 10 | NOAA | Conway and Thoning (1990) | remote |
| amt | 45.03N/68.68W | 53 | 107 | NOAA | Andrews et al. (2014) | continental |
| bao | 40.05N/105.0W | 1584 | 300 | NOAA | Andrews et al. (2014) | continental |
| lef | 45.95N/90.27W | 472 | 396 | NOAA | Andrews et al. (2014) | continental |
| sct | 33.41N/81.83W | 115 | 305 | NOAA | Andrews et al. (2014) | continental |
| snp | 38.62N/78.35W | 1008 | 17 | NOAA | Andrews et al. (2014) | mountain |
| wgc | 38.27N/121.49W | 0 | 483 | NOAA | Andrews et al. (2014) | coastal |
| wkt | 31.31N/97.33W | 251 | 457 | NOAA | Andrews et al. (2014) | continental |
| izo | 28.31N/16.50W | 2373 | 13 | AEMET | Gomez-Pelaez and Ramos (2011) | mountain |
| alt | 82.45N/62.51W | 200 | 10 | ECCC | Worthy et al. (2003) | remote |
| bck | 62.80N/116.05W | 179 | 60 | ECCC | N/A | continental |
| cby | 69.01N/105.05W | 35 | 12 | ECCC | N/A | continental |
| chl | 58.75N/94.07W | 29 | 60 | ECCC | Worthy et al. (2003) | coastal |
| cps | 49.82N/74.98W | 381 | 8 | ECCC | Worthy et al. (2003) | continental |
| egb | 44.23N/79.78W | 251 | 3 | ECCC | Worthy et al. (2003) | continental |
| esp | 49.38N/126.54W | 7 | 40 | ECCC | Worthy et al. (2003) | coastal |
| est | 51.66N/110.21W | 707 | 3 | ECCC | Worthy et al. (2003) | continental |
| etl | 54.35N/104.98W | 492 | 105 | ECCC | Worthy et al. (2003) | continental |
| fsd | 49.88N/81.57W | 210 | 40 | ECCC | Worthy et al. (2003) | continental |
| inu | 68.32N/133.53W | 113 | 10 | ECCC | Worthy et al. (2003) | continental |
| llb | 54.95N/112.45W | 540 | 10 | ECCC | Worthy et al. (2003) | continental |
| tpd | 42.62N/80.55W | 231 | 35 | ECCC | Worthy et al. (2003) | continental |
| wsa | 43.93N/60.02W | 5 | 25 | ECCC | Worthy et al. (2003) | remote |
| mnm | 24.28N/153.98E | 8 | 20 | JMA | Tsutsumi et al. (2005) | remote |





**Table A1 – continued from previous page**

| Station ID | Lat/Lon [deg] | Altitude [masl] | Intake height [masl] | Network | Reference | Type |
|---|---|---|---|---|---|---|
| ryo | 39.03N/141.82E | 260 | 20 | JMA | Tsutsumi et al. (2005) | coastal |
| yon | 24.47N/123.02E | 30 | 20 | JMA | Tsutsumi et al. (2005) | coastal |
| hdp | 40.56N/111.65W | 3351 | 17.7 | NCAR | Stephens et al. (2011) | mountain |
| nwr | 40.05N/105.59W | 3523 | 3.5 | NCAR | Stephens et al. (2011) | mountain |
| spl | 40.45N/106.73W | 3210 | 9.1 | NCAR | Stephens et al. (2011) | mountain |
| syo | 69.01S/39.59E | 14 | 8 | TU | N/A | remote |
| cpt | 34.35S/18.49E | 230 | 30 | SAWS | Brunke et al. (2004) | coastal |
| ams | 37.80S/77.54E | 55 | 20 | LSCE | Gaudry et al. (1991) | remote |
| mhd | 53.33N/9.90W | 5 | 24 | LSCE | Ramonet et al. (2010) | coastal |
| puy | 45.77N/2.97E | 1465 | 10 | LSCE | Lopez et al. (2015) | mountain |
| ces | 51.97N/4.93E | -1 | 200 | ECN | Vermeulen et al. (2011) | continental |
| wao | 52.95N/1.12E | 20 | 10 | UEA | Wilson (2013) | coastal |
| ssl | 47.92N/7.92E | 1205 | 12 | UBA-SCHAU | Schmidt et al. (2003) | mountain |
| hun | 46.95N/16.65E | 248 | 115 | HMS | Haszpra et al. (2001) | continental |
| pal | 67.97N/24.12E | 560 | 5 | FMI | Hatakka et al. (2003) | continental |
| jfj | 46.55N/7.99E | 3570 | 10 | EMPA | Schibig et al. (2015) | mountain |
| kas | 49.23N/19.98E | 1989 | 5 | AGH | Rozanski et al. (2003, 2014) | mountain |
| cgo | 40.68S/144.69E | 94 | 70 | CSIRO | Francey et al. (2003) | coastal |
| mqa | 54.50S/158.94E | 6 | 10 | CSIRO | Stavert et al. (2018) | remote |
| cya | 66.28S/110.52E | 47 | 7 | CSIRO | Francey et al. (2003) | remote |
| dec | 40.74N/0.79E | 1 | 10 | ClimaDat | Morguí et al. (2013) | coastal |
| gic | 40.35N/5.18W | 1436 | 20 | ClimaDat | Morguí et al. (2013) | mountain |
| vac | 42.88N/3.21W | 1086 | 20 | ClimaDat | Morguí et al. (2013) | mountain |
| sgc | 36.70N/5.38W | 850 | 20 | ClimaDat | Morguí et al. (2013) | continental |



**Table A2.** TCCON stations

| Station ID | Latitude/ Longitude [deg] | Altitude [m] | N data Jan | N data Jul | Organization | Reference |
|---|---|---|---|---|---|---|
| ascension01 | 7.92°S/14.33°W | 10 | 153 | 158 | MPI-BGC | Feist et al. (2014) |
| bialystok01 | 53.23°N/23.02°E | 180 | 15 | 68 | UBremen-IUP | Deutscher et al. (2014) |
| bremen01 | 53.10°N/8.85°E | 27 | 8 | 44 | UBremen-IUP | Notholt et al. (2014a) |
| darwin01 | 12.43°S/130.89°E | 30 | 34 | 264 | UOW | Griffith et al. (2014a) |
| edwards01 | 34.96°N/117.88°W | 699 | 191 | 316 | NASA | Iraci et al. (2016) |
| garmisch01 | 47.48°N/11.06°E | 740 | 33 | 90 | KIT | Sussmann and Rettinger (2014) |
| izana01 | 28.30°N/16.48°W | 2370 | 43 | 18 | AEMET/KIT | Blumenstock et al. (2014) |
| Karlsruhe | 49.10°N/8.44°E | 116 | 33 | 90 | KIT | Hase et al. (2014) |
| lamont01 | 36.60°N/97.49°W | 320 | 129 | 299 | Caltech | Wennberg et al. (2016a) |
| lauder02 | 45.04°S/169.68°E | 370 | 104 | 86 | NIWA | Sherlock et al. (2014) |
| orleans01 | 47.97°N/2.11°E | 130 | 67 | 16 | UBremen-IUP | Warneke et al. (2014) |
| parkfalls01 | 45.94°N/90.27°W | 440 | 28 | 168 | Caltech | Wennberg et al. (2014b) |
| pasadena01 | 34.14°N/118.13°W | 230 | 160 | 302 | Caltech | Wennberg et al. (2014c) |
| reunion01 | 20.90°S/55.49°E | 87 | 150 | 136 | BIRA-IASB/UR | De Mazière et al. (2014) |
| rikubetsu01 | 43.46°N/143.77°E | 30 | 21 | 9 | NIES | Morino et al. (2017a) |
| saga01 | 33.24°N/130.29°E | 7 | 30 | 30 | JAXA | Shiomi et al. (2014) |
| tsukuba02 | 36.05°N/140.12°E | 30 | 111 | 120 | NIES | Morino et al. (2014b) |
| wollongong01 | 34.41°S/150.88°E | 30 | 157 | 96 | UOW | Griffith et al. (2014b) |





**Table A3.** Organizations associated with observing stations.

| Abbreviation | Organization |
| --- | --- |
| AEMET | Izana Atmospheric Research Center, Meteorological State Agency of Spain |
| AGH | AGH University of Science and Technology, Krakzów, Poland |
| BIRA-IASB | Royal Belgian Institute for Space Aeronomy, Brussels, Belgium |
| Caltech | California Institute of Technology |
| ClimaDat | Land, atmosphere and Oceans Laboratory at the Institut Català de Ciències del Clima (2010-2016); at Institut de Ciència i Tecnologia Ambientals, Universitat Autònoma de Barcelona (since 2017) |
| CSIRO | Commonwealth Scientific and Industrial Research Organization, Oceans & Atmosphere |
| ECCC | Environment and Climate Change Canada |
| ECN | Energy Research Centre of the Netherlands |
| EMPA | Swiss Federal Laboratories for Materials Science and Technology |
| FMI | Finish Meteorological Institute |
| HMS | Hungarian Meteorological Service |
| KIT | Karlsruhe Institute of Technology |
| LSCE | Laboratoire des Sciences du Climat et de l'Environnement |
| MPI-BGC | Max Planck Institute for Biogeochemistry |
| NASA | National Aeronautics and Space Administration |
| JAXA | Japan Aerospace Exploration Agency |
| JMA | Japan Meteorological Agency |
| NIES | National Institute for Environmental Studies |
| NIWA | National Institute of Water and Atmospheric |
| NCAR | National Center For Atmospheric Research |
| NOAA | NOAA Global Monitoring Division |
| SAWS | South African Weather Service |
| TU | Tohoku University |
| UBA-SCHAU | Umweltbundesamt, Station Schauinsland |
| UBremen-IUP | Institute of Environmental Physics, Universität Bremen |
| UEA | University of East Anglia |
| UHEI-IUP | University of Heidelberg, Institut fuer Umweltphysik |
| UOW | University of Wollongong |
| UR | Université de La Réunion |





*Author contributions.* The simulations were performed by A.Agustí-Panareda; the coding of the mass fixer required for the high resolution transport in the IFS was done by M. Diamantakis; the concept and ideas to design the high resolution simulations were devised by F. Chevallier, A. Agustí-Panareda, M. Diamantakis, S. Massart, J. Muñoz-Sabater in discussion with R. Engelen and V.-H. Peuch; R. Law, Z. Loh, J.A. Morguí, R. Curcoll provided additional observations at crucial sites and guidance on the evaluation of the simulations; C. Roehl and D. Wunch provided data and input on the interpretation of the model evaluation at the TCCON site of Pasadena; the $CO_2$ validation tools have been developed by S. Massart and A. Agustí-Panareda; the manuscript was prepared by A. Agustí-Panareda with input and feedback from M. Diamantakis, S. Massart, F. Chevallier, J. Muñoz-Sabater, J. Barré, R. Engelen, B. Langerock, R. Law, Z. Loh, J.A. Morguí, M. Parrington, V.-H. Peuch, M. Ramonet, C. Roehl, A.T. Vermeulen, T. Warneke and D. Wunch.

*Competing interests.* The authors declare that they have no conflict of interest.

*Acknowledgements.* This research was generated using Copernicus Atmosphere Monitoring Service Information [2018]. Anna Agustí-Panareda has been partly funded by the CHE project. The CHE project has received funding from the European Union's Horizon 2020 research and innovation programme under grant agreement No 776186. Frédéric Chevallier received funding from the Copernicus Atmosphere Monitoring Service, implemented by the European Centre for Medium-Range Weather Forecasts (ECMWF) on behalf of the European Commission. Thanks to all the station PIs from the numerous individual stations and networks like NOAA, ICOS, AEMET, AGH, CSIRO, ECCC, ECN, EMPA,FMI, HMS, LSCE, NCAR, JMA, MPI-BGC, NIWA, SAWS, TU, UBA-SCHAU, UEA, UHEI-IUP, UR, UBremen-IUP, BIRA-IASB, Caltech, KIT, NASA, JAXA, NIES, UOW that contributed their $CO_2$ observations to the cooperative GLOBALVIEWplus ObsPack (2015) product and TCCON which are fundamental for the evaluation of the model simulations. The ClimaDat Network has received funding from "la Caixa" Foundation, under agreement 2010-002624. We are grateful to many colleagues at ECMWF for their support and fruitful discussions, particularly to Gabor Radnoti, Thomas Haiden and Martin Janouseck for their technical support in the evaluation of the winds, Johannes Flemming and Zak Kipling for their support in the implementation of the additional $CO_2$ tracers in the IFS model, Sylvie Malardel for her support and discussions on the general aspects of atmospheric tracer modelling in the IFS, Gianpaolo Balsamo, Souhail Boussetta, Zak Kipling and Johannes Flemming for their technical support on the implementation of a bug fix in the CTESSEL model of biogenic emissions. Many thanks to Paul Wennberg (Caltech) for his advice on the use of the TCCON data, Martin Krol (Wageningen University) for his suggestions on the evaluation of the daily maximum $CO_2$ and Britton Stephens (NCAR) for his comments on the mountain site evaluation which helped improve the description of the vertical sampling strategy and to emphasize the importance of high resolution at mountain sites.





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
