# Peer review of "Modelling CO2 weather – why horizontal resolution matters"

_Atmospheric Chemistry and Physics, 2019_

## Referee Comment (RC1) · Anonymous Referee #1 · 17 Apr 2019

This manuscript considers the benefit of high horizontal resolution for the simulation of CO2 transport. The article uses the CAMS model at a variety of resolutions with grid sizes ranging from 80 to 9 km. The performance of the models validated against CO2 surface and column observations from TCCON reveals improved performance with higher horizontal resolution. In addition, the forecast error is smaller with higher resolution, at all forecast ranges (from 1 to 10 days). Finally, the high resolution model is used to quantify subgrid scale variability of surface and column mean CO2 for a coarser (1x1 degree) grid. Overall, the article is well written, with a clear and well-supported message. The nuances of the issue (such as the nonlinearity of forecast error growth with forecast range) are also discussed. The question of the optimal horizontal resolution needed for CO2 transport depends on the goal of the assimilation

problem and the observations used to constrain the inversion system. With the rapid expansion of the global observation network and the desire to resolve fluxes at finer spatial scales, the flux estimation question is shifting to one of anthropogenic attribution so the question of model error sensitivity to horizontal resolution is both important and timely.

Minor comments

1. P2, L2-4: These 2 sentences seem contradictory. Please resolve the conflict.

2. P3, L6: "being" should be "whether"

3. P4, L8: Figure 3 is referenced before Figure 2 (P6, L21), therefore please reorder these figures.

4. P4, L15: Table A3 is referenced before Table A2. Please reverse the order of these Tables.

5. P5, L17: "semi-lagrangian" needs a capital L

6. P6, L1- 4: Do the net sources and sinks of biogenic fluxes have the same value across horizontal resolutions? Are the global net fluxes the same for all resolutions? An example showing biogenic fluxes from different resolutions would be helpful in this regard.

7. P6, L14: "Semi-Implicit Semi-" doesn't need any capitals.

8. P8, L12-13: "Since most low resolution models used in atmospheric inversions tend to use the model sampling ASL at mountain sites. . ." Please add some references here to back up this statement.

9. P9, section 3.1 and Fig. 4: Care must be taken when interpreting meteorological forecasts at 1000 hPa because this level frequently requires extrapolation (below mountains and at locations with surface pressure below 1000 hPa). Some caveats regarding the use of this level should be mentioned.

[Figure]

10. P9, L15-16: Diagnostics computed with observations at screen level are mentioned but not shown. It would be worth showing these figures because of the issues with observations at 1000 hPa mentioned in point 9, and because there are far more observations from surface stations than there are radiosonde observations at 1000 hPa.

11. P9, L32: "reflect on" should be "reflect"

12. P10, L2: "Figs. 5c and 5d" should be "Figs. 6a and 6b"

13. LP11, L17: "summer (winter)". Should this be "boreal summer (winter)"? There may be similar issues occurring elsewhere, for example on P14, L15. Please review the entire manuscript to ensure clarity when discussing seasons in regard to global results.

14. P13, L13: "an" should be "and".

15. P14, L33 and P15 L1: please keep consistency between "sec." and "section".

16. P16, L26: "at least 4 km". Do you mean "at most 4 km"? Also it would be clearer to talk about grid spacing rather than resolution.

17. Tables S3, S4: It would be helpful for the reader to provide the difference in bias and possibly also for standard error, as it is for RMSE. Also, it would be better to order the stations by latitude rather than by name, to better see if there are any patterns with respect to latitude.

---

## Referee Comment (RC2) · Anonymous Referee #2 · 22 Apr 2019

This study demonstrates the advantage of using a high-resolution model by taking into account of the effect of atmospheric transport, flux variations, and topography. An ensemble of ground-based continuous measurements all over the globe that includes CO2 surface network and TCCON (column) are used to derive model errors. The model errors that are arisen only due to fine-scale horizontal variability are deduced by using single modeling system (CAMS) at various horizontal grid sizes, but maintaining the vertical resolution and using the same set of flux inventories. The local flux influence and its dependence on resolution are investigated by switching off the CO2 surface fluxes. Noteworthy is that the study also shows the importance of representing local gradients of CO2 fluxes in urban regions and during night-times for reducing the atmospheric CO2 representativeness error. The dependence of forecast skill on hori-

zontal resolution and extent of the forecasting period is discussed. The topic is of very high interest to the ACP community, particularly relevant for atmospheric inverse modelers who aim to retrieve CO2 sources and sinks at regional scales. The manuscript is well written; the analyses are conducted carefully and the results are presented in a logical order with appropriate support and interpretation. I recommend this paper for publication in ACP, after addressing the comments below.

Comments:

Impact of horizontal resolution on representation errors (P11-12, Fig. 11): I see data gaps in Fig. 11. In my understanding, the representation errors are calculated using the standard deviations of modeled concentrations at a fine scale (9km interpolated to 0.1 degree) within the global grid boxes of 1 degree $\times$ 1 degree. In that case, I can't understand why such data gaps exist? Had any filtering been adopted? Please clarify. Also, it would be very helpful if the manuscript includes the monthly averaged modeled simulations at 9 and 80 km resolutions (spatial plot) for surface and column concentrations. I would suggest authors include those plots, allowing the reader to do the visual comparison in terms of statistical (as done in Fig.11) and model-predicted (9 km vs. 80 km) sub-grid variability.

Table 3 and Fig. 5: Why there exists difference (in magnitude) between the standard deviation of inter-station RMSE (sigma-RMSE) given in Table 3 ((in brackets and in bold, last column) and those given in Fig. 5 (a) & (b)? I assume that the authors used "All stations" in January and July for these calculations.

Fig. 7 (b): XCO2 daily min vs. daily mean/max in July. It's rather surprising to see the high RMSE values for daily min. What caused RMSE (daily min) to be almost doubled compared to RMSE (daily mean) and RMSE (daily max, nighttime?), given that RMSE (hourly) doesn't show this high value?

Minor comments:

[Figure]
* * *
Interactive
comment

Fig. 1: In Fig. Caption, please indicate the model's resolution used.

Fig. 5, 6 & 7: In Fig. Caption, the standard deviation of R is not mentioned though it is given in the plot. You may please rewrite as: "The standard deviation of the plotted variable from each station is shown…"

Table 2: Since there is no change in flux datasets used for different experiments, please remove the last column and indicate details of $CO_2$ fluxes in the figure caption.

---

## Author Comment (AC1) · 3 May 2019

The authors would like to thank the reviewers for the comments which have been addressed below and have contributed to improve the clarity of the manuscript. All the corrections in the revised manuscript have been highlighted in blue and bold phase (see Supplement file including revised manuscript and supplement).

**Minor comments**

- 1. P2, L2-4: These 2 sentences seem contradictory. Please resolve the conflict.

The last sentence has been re-written to clarify the difficulty in extrapolating the results to higher resolutions: "It is clear from the results that an additional increase in resolution might reduce errors even further. However, the horizontal resolution sensitivity tests indicate that the change in the $CO_2$ and wind modelling error with resolution is not linear, making it difficult to **quantify** the **improvement** beyond the tested resolutions."

- **2. P3, L6: "being" should be "whether"**

  Done.

- **3. P4, L8: Figure 3 is referenced before Figure 2 (P6, L21), therefore please reorder these figures.**

  Done.

- **4. P4, L15: Table A3 is referenced before Table A2. Please reverse the order of these Tables.**

  Done.

- **5. P5, L17: "semi-lagrangian" needs a capital L**

  Done.

- **6. P6, L1- 4: Do the net sources and sinks of biogenic fluxes have the same value across horizontal resolutions? Are the global net fluxes the same for all resolutions? An example showing biogenic fluxes from different resolutions would be helpful in this regard.**

  A figure with the monthly mean NEE from the 9km-EXP and 80km-EXP simulations has been added in the Supplement (see Fig. S1). The diffence in their global budget is less than 1%. This has been mentioned in the revised manuscript.

- **7. P6, L14: "Semi-Implicit Semi-" doesn't need any capitals.**

  Done.

- **8. P8, L12-13: "Since most low resolution models used in atmospheric inversions tend to use the model sampling ASL at mountain sites. . ." Please add some references here to back up this statement.**

  Done.

- **9. P9, section 3.1 and Fig. 4: Care must be taken when interpreting meteorologi- cal forecasts at 1000 hPa because this level frequently requires extrapolation (below mountains and at locations with surface pressure below 1000 hPa). Some caveats regarding the use of this level should be mentioned.**

  A cautionary note has been added to the revised manuscript (see Fig. 4 caption): "Note that the number of data at 1000-hPa level might be lower than the other levels as observations will be missing when the surface pressure is lower than 1000 hPa."

- **10. P9, L15-16: Diagnostics computed with observations at screen level are men- tioned but not shown. It would be worth showing these figures because of the issues with observations at 1000 hPa mentioned in point 9, and because there are far more observations from surface stations than there are radiosonde observations at 1000 hPa.**

  The global RMSE and global bias reduction are shown in the revised manuscript with consistent values to the 1000hPa winds.

- **11. P9, L32: "reflect on" should be "reflect"**

  Done.

- **12. P10, L2: "Figs. 5c and 5d" should be "Figs. 6a and 6b"**

  Done.

- **13. LP11, L17: "summer (winter)". Should this be "boreal summer (winter)"? There may be similar issues occurring elsewhere, for example on P14, L15. Please review the entire manuscript to ensure clarity when discussing seasons in regard to global results.**

  Done. When summer/winter is used it is valid for either both hemispheres or a specific site. When winter/summer refer to the January/July plots, then "boreal" is used.

- **14. P13, L13: "an" should be "and".**

  Done.

- **15. P14, L33 and P15 L1: please keep consistency between "sec." and "section".**

  Done.

- **16. P16, L26: "at least 4 km". Do you mean "at most 4 km"? Also it would be clearer to talk about grid spacing rather than resolution.**

  The sentence has been re-written to explain that 4km is the minimum horizontal resolution required. A clarification of the equivalence between horizontal resolution and model grid spacing has been included in the introduction.

- **17. Tables S3, S4: It would be helpful for the reader to provide the difference in bias and possibly also for standard error, as it is for RMSE. Also, it would be better to order the stations by latitude rather than by name, to better see if there are any patterns with respect to latitude**

  Done.

Please also note the supplement to this comment:
https://www.atmos-chem-phys-discuss.net/acp-2019-177/acp-2019-177-AC1-supplement.pdf

---

## Author Comment (AC2) · 3 May 2019

The authors would like to thank the reviewer for the comments which have been addressed below and have contributed to improve the clarity of the manuscript. All the corrections in the revised manuscript have been highlighted in blue and bold phase (see Supplement file including both revised manuscript and supplement).

**General comments**

- **Impact of horizontal resolution on representation errors (P11-12, Fig. 11): I see data gaps in Fig. 11. In my understanding, the representation errors**

[Figure]

**are calculated using the standard deviations of modeled concentrations at a fine scale (9km interpolated to 0.1 degree) within the global grid boxes of 1 degree × 1 degree. In that case, I can't understand why such data gaps exist? Had any filtering been adopted? Please clarify.**

The grey area which appears as data gaps shows the regions where $\sigma$ is less than the threshold value of 1ppm for surface $CO_2$ and 0.1ppm for $XCO_2$. This has now been clarified in all the relevant figure captions.

- **Also, it would be very helpful if the manuscript includes the monthly averaged modeled simulations at 9 and 80 km resolutions (spatial plot) for surface and column concentrations. I would suggest authors include those plots, allowing the reader to do the visual comparison in terms of statistical (as done in Fig.11) and model-predicted (9 km vs. 80 km) sub-grid variability.**

Monthly mean plots of surface $CO_2$ and $XCO_2$ have been included in the Supplement (see Figs S6 and S7) to provide a visual illustration of the small-scale variability associated with the 9km-EXP simulation compared to the 80km-EXP simulation.

- **Table 3 and Fig. 5: Why there exists difference (in magnitude) between the standard deviation of inter-station RMSE (sigma-RMSE) given in Table 3 ((in brackets and in bold, last column) and those given in Fig. 5 (a) and (b)? I assume that the authors used "All stations" in January and July for these calculations.**

Table 3 had not been updated when the number of observations was slightly changed (e.g. only the top level at the tower sites is used as listed in Table A1). This has now been corrected in the revised manuscript, so that Table 3 and Fig. 5 are consistent.

- **Fig. 7 (b): XCO2 daily min vs. daily mean/max in July. It's rather surprising to see the high RMSE values for daily min. What caused RMSE (daily min) to be almost doubled compared to RMSE (daily mean) and RMSE (daily max, nighttime?), given that RMSE (hourly) doesn't show this high value?**

  The XCO2 from TCCON is only available during daytime. This has been clarified in the revised manuscript. The CO2 daily minimum in July (boreal summer) is more uncertain than the daily maximum because it is controlled by the dominant biogenic fluxes associated with photosynthesis (i.e. negative XCO2 anomalies); whereas in January most sites (i.e. those in NH) are dominated by respiration, affecting the daily maximum variability and its RMSE. This has also been clarified in the revised manuscript.

**Minor comments**

- **Fig. 1: In Fig. Caption, please indicate the model's resolution used.**

  Done.

- **Fig. 5, 6 and 7: In Fig. Caption, the standard deviation of R is not mentioned though it is given in the plot. You may please rewrite as: "The standard deviation of the plotted variable from each station is shown. . ."**

  Done.

- **Table 2: Since there is no change in flux datasets used for different experiments, please remove the last column and indicate details of CO2 fluxes in the figure caption.**

  Done.

Please also note the supplement to this comment:
https://www.atmos-chem-phys-discuss.net/acp-2019-177/acp-2019-177-AC2-
supplement.pdf

**Supplement:**

[revised manuscript text omitted]

**Figure S11.** Same as Fig. **??** in July.

[Figure]

**Figure S12.** Modelled atmospheric $CO_2$ profile [ppm] showing 137 model levels at Pasadena from the 9-km resolution simulation and the 80-km resolution simulation. The horizontal black line shows the level approximately corresponding to 1000 m above the surface. In winter, the high $CO_2$ values are trapped below 1000 m in the boundary layer, except for certain synoptic episodes where boundary layer is ventilated as seen by high values (larger than 420 ppm) crossing the 1000 m line. While in summer the sea breeze circulation ventilates the boundary layer on a daily basis.

Table S1: January statistics of atmospheric $CO_2$ [ppm] from 9km-EXP and 80km-EXP simulations with respect to continuous in situ stations (surface and tower). The differences between the two simulations (9km-EXP – 80km-EXP) are shown in the last three columns. The location and reference of each station can be found in Tab. A1.

| Station | Bias 9kmEXP | Bias 80kmEXP | STDE 9kmEXP | STDE 80kmEXP | RMSE 9kmEXP | RMSE 80kmEXP | N data | Δ STDE | Δ \|Bias\| | Δ RMSE |
|---|---|---|---|---|---|---|---|---|---|---|
| alt | -2.14 | -2.09 | 0.61 | 0.61 | 2.23 | 2.18 | 679 | 0.05 | 0.00 | 0.05 |
| brw | -1.45 | -1.55 | 1.19 | 1.14 | 1.88 | 1.92 | 727 | -0.10 | 0.05 | -0.04 |
| cby | -1.51 | -1.30 | 0.58 | 0.56 | 1.62 | 1.42 | 298 | 0.21 | 0.02 | 0.2 |
| inu | 0.50 | 0.36 | 2.04 | 2.45 | 2.11 | 2.47 | 702 | 0.14 | -0.41 | -0.36 |
| pal-nonlocal | -0.89 | 0.32 | 2.13 | 3.44 | 2.31 | 3.46 | 595 | 0.57 | -1.31 | -1.15 |
| bck | 0.93 | 1.21 | 1.45 | 1.37 | 1.73 | 1.83 | 768 | -0.28 | 0.08 | -0.1 |
| chl | 0.50 | 0.68 | 1.70 | 1.91 | 1.77 | 2.02 | 465 | 0.18 | -0.21 | -0.25 |
| llb | -1.77 | -1.59 | 3.19 | 3.32- | 3.65 | 3.69 | 699 | 0.18 | -0.13 | -0.04 |
| etl | -0.67 | -0.17 | 1.53 | 1.71 | 1.68 | 1.72 | 695 | 0.5 | -0.17 | -0.04 |
| mhd | -1.11 | 2.34 | 4.75 | 1.03 | 1.52 | 5.30 | 759 | -1.23 | 3.72 | -3.78 |
| wao | -0.68 | -0.01 | 3.24 | 3.26 | 3.31 | 3.26 | 92 | 0.67 | -0.02 | 0.05 |
| ces-200magl | -2.39 | -1.27 | 4.41 | 4.71 | 5.02 | 4.88 | 679 | 1.12 | -0.30 | 0.14 |
| est | -1.20 | -1.08 | 2.09 | 1.93 | 2.41 | 2.21 | 762 | 0.12 | 0.16 | 0.2 |
| fsd | -0.73 | -0.40 | 1.24 | 1.29 | 1.44 | 1.35 | 768 | 0.33 | -0.05 | 0.09 |
| cps | -0.58 | 0.06 | 1.32 | 1.49 | 1.44 | 1.49 | 697 | 0.52 | -0.17 | -0.05 |
| esp | 1.01 | 4.43 | 3.71 | 4.93 | 3.84 | 6.62 | 753 | -3.42 | -1.22 | -2.78 |
| kas | 0.67 | 8.45 | 4.39 | 6.57 | 4.44 | 10.71 | 503 | -7.78 | -2.18 | -6.27 |
| ssl | 3.21 | 18.72 | 4.87 | 15.00 | 5.83 | 23.99 | 739 | -15.51 | -10.13 | -18.16 |
| hun-115magl | -6.58 | -2.68 | 5.52 | 5.64 | 8.59 | 6.24 | 751 | 3.90 | -0.12 | 2.35 |
| jfj | 0.08 | 12.47 | 2.53 | 9.29 | 2.53 | 15.55 | 720 | -12.39 | -6.76 | -13.02 |
| lef-396magl | -0.78 | -0.49 | 1.47 | 1.51 | 1.67 | 1.59 | 765 | 0.29 | -0.04 | 0.08 |
| puy | 2.39 | 6.11 | 3.91 | 8.29 | 4.58 | 10.30 | 752 | -3.72 | -4.38 | -5.72 |
| amt-107magl | 0.01 | -0.50 | 2.68 | 2.77 | 2.68 | 2.81 | 741 | -0.49 | -0.09 | -0.13 |
| egb | -1.13 | -1.66 | 5.14 | 5.33 | 5.26 | 5.58 | 726 | -0.53 | -0.19 | -0.32 |
| wsa | -1.24 | -0.82 | 1.22 | 1.43 | 1.74 | 1.65 | 766 | 0.38 | -0.21 | 0.09 |
| vac | -0.13 | 1.82 | 1.10 | 1.82 | 1.10 | 2.28 | 161 | -1.69 | -0.72 | -1.82 |
| tpd | -0.01 | 0.81 | 3.11 | 3.20 | 3.11 | 3.30 | 768 | -0.8 | -0.09 | -0.19 |

| Station | Bias 9kmEXP | Bias 80kmEXP | STDE 9kmEXP | STDE 80kmEXP | RMSE 9kmEXP | RMSE 80kmEXP | N data | Δ STDE | Δ \|Bias\| | Δ RMSE |
|---|---|---|---|---|---|---|---|---|---|---|
| dec | 11.13 | 7.43 | 11.42 | 7.31 | 15.95 | 10.42 | 588 | 3.70 | 4.11 | 5.53 |
| hdp | 1.48 | 16.83 | 2.73 | 10.25 | 3.10 | 19.71 | 668 | -15.35 | -7.52 | -16.61 |
| spl | 2.28 | 2.72 | 3.23 | 3.50 | 3.95 | 4.43 | 682 | -0.44 | -0.27 | -0.48 |
| gic | -1.88 | 1.69 | 5.28 | 4.43 | 5.60 | 4.74 | 765 | 0.19 | 0.85 | 0.86 |
| nwr | 0.76 | 1.56 | 1.45 | 3.40 | 1.64 | 3.74 | 730 | -0.80 | -1.95 | -2.1 |
| bao-300magl | 0.34 | -2.20 | 9.43 | 8.19 | 9.43 | 8.48 | 760 | -1.86 | 1.24 | 0.95 |
| ryo | 3.05 | 3.88 | 4.99 | 6.05 | 5.84 | 7.19 | 432 | -0.83 | -1.06 | -1.35 |
| snp-17magl | 3.05 | 9.66 | 3.97 | 10.87 | 5.01 | 14.54 | 768 | -6.61 | -6.90 | -9.53 |
| wgc-483magl | -0.58 | -0.60 | 4.92 | 5.71 | 4.95 | 5.74 | 768 | -0.02 | -0.79 | -0.79 |
| sgc | 1.31 | 10.31 | 5.61 | 9.62 | 5.76 | 14.10 | 652 | -9.00 | -4.01 | -8.34 |
| sct-305magl | -0.13 | 0.42 | 3.61 | 3.83 | 3.62 | 3.85 | 762 | -0.29 | -0.22 | -0.23 |
| wkt-457magl | 0.06 | 0.22 | 2.34 | 2.38 | 2.34 | 2.39 | 733 | -0.16 | 0.04 | -0.05 |
| izo | 0.01 | 0.63 | 2.80 | 0.98 | 2.80 | 1.16 | 722 | -0.62 | 1.82 | 1.64 |
| yon | -0.40 | -0.62 | 1.22 | 1.43 | 1.28 | 1.56 | 579 | -0.22 | -0.21 | -0.28 |
| mnm | -0.34 | -0.25 | 0.77 | 0.71 | 0.84 | 0.76 | 680 | 0.09 | 0.06 | 0.08 |
| mlo | -0.35 | 0.68 | 0.78 | 1.05 | 0.85 | 1.25 | 736 | -0.33 | -0.27 | -0.4 |
| smo | -1.10 | -0.81 | 0.93 | 0.97 | 1.44 | 1.26 | 683 | 0.29 | -0.04 | 0.18 |
| cpt-marine | -1.11 | 1.86 | 0.60 | 6.02 | 1.26 | 6.30 | 618 | -0.75 | -5.42 | -5.04 |
| ams | -1.20 | -1.27 | 0.26 | 0.27 | 1.22 | 1.30 | 116 | -0.07 | -0.01 | -0.08 |
| cgo | -0.69 | -1.39 | 2.46 | 4.25 | 2.56 | 4.47 | 768 | -0.70 | -1.79 | -1.91 |
| mqa | -1.11 | -1.26 | 0.65 | 0.66 | 1.29 | 1.43 | 618 | -0.15 | -0.01 | -0.14 |
| cya | -1.14 | -1.14 | 0.36 | 0.36 | 1.19 | 1.19 | 693 | 0.00 | 0.00 | 0.00 |
| syo | -1.09 | -1.15 | 0.14 | 0.15 | 1.10 | 1.16 | 32 | -0.06 | -0.01 | -0.06 |
| spo | -1.10 | -1.10 | 0.18 | 0.19 | 1.12 | 1.12 | 736 | 0.00 | -0.01 | 0.00 |

Table S2: July statistics of atmospheric $CO_2$ [ppm] from 9km-EXP and 80km-EXP simulations with respect to continuous in situ stations (surface and tower). The differences between the two simulations (9km-EXP – 80km-EXP) are shown in the last three columns. The location and reference of each station can be found in Tab. A1.

| Station | Bias 9kmEXP | Bias 80kmEXP | STDE 9kmEXP | STDE 80kmEXP | RMSE 9kmEXP | RMSE 80kmEXP | N data | Δ STDE | Δ \|Bias\| | Δ RMSE |
|---|---|---|---|---|---|---|---|---|---|---|
| alt | -0.93 | -1.36 | 1.05 | 1.17 | 1.40 | 1.80 | 623 | -0.43 | -0.12 | -0.4 |
| brw | -0.85 | -0.68 | 2.06 | 2.20 | 2.23 | 2.31 | 738 | 0.17 | -0.14 | -0.08 |
| cby | -0.67 | -1.83 | 3.07 | 3.50 | 3.15 | 3.95 | 754 | -1.16 | -0.43 | -0.8 |
| inu | -1.40 | -2.54 | 3.98 | 5.07 | 4.22 | 5.67 | 765 | -1.14 | -1.09 | -1.45 |
| pal-nonlocal | 2.03 | 4.40 | 6.13 | 10.86 | 6.45 | 11.72 | 345 | -2.37 | -4.73 | -5.27 |
| bck | 10.36 | 34.84 | 38.58 | 79.33 | 39.95 | 86.65 | 757 | -24.48 | -40.75 | -46.7 |
| chl | -0.09 | -0.77 | 4.45 | 4.61 | 4.45 | 4.67 | 768 | -0.68 | -0.16 | -0.22 |
| llb | -10.09 | -7.88 | 14.30 | 13.39 | 17.50 | 15.53 | 352 | 2.21 | 0.91 | 1.97 |
| etl | -3.59 | -4.90 | 7.02 | 7.48 | 7.88 | 8.94 | 549 | -1.31 | -0.46 | -1.06 |
| mhd | -2.27 | -0.40 | 5.63 | 6.52 | 6.07 | 6.53 | 703 | 1.87 | -0.89 | -0.46 |
| wao | -4.01 | -3.44 | 8.33 | 7.12 | 9.24 | 7.91 | 568 | 0.57 | 1.21 | 1.33 |
| ces-200magl | -3.49 | -2.93 | 7.76 | 7.97 | 8.51 | 8.50 | 668 | 0.56 | -0.21 | 0.01 |
| est | 0.35 | 0.50 | 8.62 | 9.37 | 8.63 | 9.38 | 79 | -0.15 | -0.75 | -0.75 |
| fsd | -3.51 | -4.59 | 8.96 | 9.23 | 9.62 | 10.31 | 768 | -1.08 | -0.27 | -0.69 |
| cps | -2.98 | -3.85 | 7.03 | 7.52 | 7.64 | 8.45 | 760 | -0.87 | -0.49 | -0.81 |
| esp | 0.28 | -6.53 | 5.69 | 10.09 | 5.70 | 12.01 | 318 | -6.25 | -4.40 | -6.31 |
| kas | -1.01 | 7.42 | 4.17 | 15.93 | 4.29 | 17.57 | 558 | -6.41 | -11.76 | -13.28 |
| ssl | -0.11 | 9.63 | 8.99 | 18.56 | 8.99 | 20.91 | 761 | -9.52 | -9.57 | -11.92 |
| hun-115magl | -6.61 | -5.61 | 7.87 | 7.43 | 10.28 | 9.32 | 768 | 1.0 | 0.44 | 0.96 |
| jfj | -5.23 | -5.48 | 3.60 | 10.60 | 6.35 | 11.93 | 109 | -0.25 | -7.0 | -5.58 |
| lef-396magl | 3.88 | 2.53 | 6.22 | 6.05 | 7.33 | 6.56 | 744 | 1.35 | 0.17 | 0.77 |
| puy | 0.75 | 4.88 | 7.19 | 12.36 | 7.23 | 13.29 | 752 | -4.13 | -5.17 | -6.06 |
| amt-107magl | 2.60 | -0.94 | 8.24 | 7.95 | 8.64 | 8.00 | 768 | 1.66 | 0.29 | 0.64 |
| egb | -1.24 | -6.52 | 13.31 | 15.61 | 13.37 | 16.92 | 632 | -5.28 | -2.30 | -3.55 |
| wsa | 0.95 | 0.41 | 4.66 | 5.60 | 4.76 | 5.62 | 768 | 0.54 | -0.94 | -0.86 |
| vac | 2.85 | 6.98 | 5.22 | 12.04 | 5.95 | 13.91 | 764 | -4.13 | -6.82 | -7.96 |
| tpd | -1.20 | -2.44 | 14.31 | 13.12 | 14.37 | 13.34 | 767 | -1.24 | 1.19 | 1.03 |

| Station | Bias 9kmEXP | Bias 80kmEXP | STDE 9kmEXP | STDE 80kmEXP | RMSE 9kmEXP | RMSE 80kmEXP | N data | △ STDE | △ \|Bias\| | △ RMSE |
|---|---|---|---|---|---|---|---|---|---|---|
| dec | 7.78 | 11.01 | 10.37 | 11.97 | 12.96 | 16.26 | 619 | -3.23 | -1.60 | -3.3 |
| hdp | 4.11 | 27.16 | 4.36 | 25.67 | 5.99 | 37.37 | 551 | -23.05 | -21.31 | -31.38 |
| spl | 8.73 | 20.16 | 6.34 | 16.93 | 10.79 | 26.32 | 493 | -11.43 | -10.59 | -15.53 |
| gic | -10.88 | -6.14 | 17.13 | 14.08 | 20.30 | 15.36 | 147 | 4.74 | 3.05 | 4.94 |
| nwr | 3.63 | 11.03 | 3.68 | 15.20 | 5.17 | 18.78 | 399 | -7.40 | -11.52 | -13.61 |
| bao-300magl | 1.05 | -1.43 | 5.69 | 6.55 | 5.79 | 6.70 | 743 | -0.38 | -0.86 | -0.91 |
| ryo | 18.51 | 10.77 | 27.89 | 17.28 | 33.48 | 20.36 | 170 | 7.74 | 10.61 | 13.12 |
| snp-17magl | 24.15 | 37.81 | 16.55 | 30.11 | 29.28 | 48.33 | 768 | -13.66 | -13.56 | -19.05 |
| wgc-483magl | 1.57 | 1.37 | 2.75 | 2.81 | 3.17 | 3.13 | 384 | 0.20 | -0.06 | 0.04 |
| sgc | 5.74 | 14.61 | 5.71 | 12.54 | 8.09 | 19.25 | 719 | -8.87 | -6.83 | -11.16 |
| sct-305magl | 3.90 | 4.21 | 7.82 | 7.31 | 8.73 | 8.43 | 767 | -0.31 | 0.51 | 0.3 |
| wkt-457magl | 4.75 | 4.90 | 4.32 | 3.93 | 6.42 | 6.28 | 666 | -0.15 | 0.39 | 0.14 |
| izo | 4.65 | -1.84 | 3.82 | 2.22 | 6.01 | 2.88 | 746 | 2.81 | 1.60 | 3.13 |
| yon | 0.61 | 0.26 | 1.98 | 1.58 | 2.07 | 1.60 | 522 | 0.35 | 0.40 | 0.47 |
| mnm | 0.33 | 0.27 | 0.98 | 0.97 | 1.04 | 1.00 | 647 | 0.06 | 0.01 | 0.04 |
| mlo | 0.83 | -0.52 | 1.22 | 1.60 | 1.47 | 1.68 | 612 | 0.31 | -0.38 | -0.21 |
| smo | -0.26 | -0.34 | 0.80 | 0.87 | 0.84 | 0.93 | 695 | -0.08 | -0.07 | -0.09 |
| cpt-36-marine | -0.12 | -0.82 | 0.93 | 5.91 | 0.94 | 5.97 | 536 | -0.7 | -4.98 | -5.03 |
| ams | -0.90 | -1.03 | 0.28 | 0.29 | 0.94 | 1.07 | 306 | -0.13 | -0.01 | -0.13 |
| cgo | -0.55 | -0.41 | 1.56 | 2.42 | 1.66 | 2.45 | 758 | 0.14 | -0.86 | -0.79 |
| mqa | -0.84 | -0.94 | 0.40 | 0.40 | 0.93 | 1.02 | 692 | -0.10 | 0.0 | -0.09 |
| cya | -0.95 | -1.01 | 0.29 | 0.29 | 0.99 | 1.05 | 760 | -0.06 | 0.0 | -0.06 |
| syo | -0.92 | -0.97 | 0.14 | 0.13 | 0.93 | 0.98 | 32 | -0.05 | 0.01 | -0.05 |
| spo | -0.83 | -0.88 | 0.16 | 0.15 | 0.85 | 0.89 | 737 | -0.05 | 0.01 | -0.04 |

Table S3: January statistics of $XCO_2$ [ppm] from 9km-EXP and 80km-EXP simulations with respect to TCCON stations. The differences between the two simulations (9km-EXP – 80km-EXP) are shown in the last three columns. The location of the station and their associated reference are provided in Tab A3.

| Station | Bias 9kmEXP | Bias 80kmEXP | STDE 9kmEXP | STDE 80kmEXP | RMSE 9kmEXP | RMSE 80kmEXP | N data | $\triangle$ STDE | $\triangle$ |Bias| | $\triangle$ RMSE |
|---|---|---|---|---|---|---|---|---|---|---|
| bialystok01 | -1.28 | -1.20 | 0.29 | 0.33 | 1.32 | 1.24 | 15 | 0.08 | -0.04 | 0.08 |
| bremen01 | 0.22 | 0.12 | 0.73 | 0.70 | 0.77 | 0.71 | 8 | 0.10 | 0.03 | -0.06 |
| karlsruhe01 | 0.11 | -0.12 | 0.69 | 0.54 | 0.70 | 0.55 | 33 | -0.01 | 0.15 | 0.15 |
| orleans01 | -0.25 | -0.26 | 0.45 | 0.48 | 0.52 | 0.54 | 67 | -0.01 | -0.03 | -0.02 |
| garmisch01 | -0.53 | -0.73 | 0.55 | 0.60 | 0.76 | 0.94 | 33 | -0.20 | -0.05 | -0.18 |
| parkfalls01 | -1.42 | -1.38 | 0.37 | 0.36 | 1.46 | 1.42 | 28 | 0.04 | 0.01 | 0.04 |
| rikubetsu01 | -1.74 | -1.74 | 0.14 | 0.17 | 1.74 | 1.75 | 21 | 0.00 | -0.03 | -0.01 |
| lamont01 | -0.98 | -1.03 | 0.64 | 0.66 | 1.17 | 1.22 | 129 | -0.05 | -0.02 | -0.05 |
| tsukuba02 | 0.37 | 0.36 | 1.02 | 0.97 | 1.08 | 1.03 | 111 | 0.01 | 0.05 | 0.05 |
| edwards01 | 0.42 | 0.05 | 0.47 | 0.64 | 0.63 | 0.65 | 191 | 0.35 | -0.17 | -0.02 |
| pasadena01 | 0.32 | 2.03 | 1.24 | 2.68 | 1.28 | 3.36 | 160 | -1.71 | -1.44 | -2.08 |
| saga01 | -1.30 | -1.36 | 0.78 | 0.64 | 1.52 | 1.50 | 30 | -0.06 | 0.14 | 0.02 |
| izana01 | -0.85 | -0.36 | 0.39 | 0.29 | 0.93 | 0.46 | 18 | 0.49 | 0.10 | 0.47 |
| ascension01 | 0.69 | 0.89 | 0.71 | 0.72 | 0.99 | 1.15 | 153 | -0.20 | -0.01 | -0.16 |
| darwin01 | -1.04 | -1.17 | 0.49 | 0.76 | 1.15 | 1.39 | 34 | -0.13 | -0.27 | -0.24 |
| reunion01 | -0.10 | 0.01 | 0.34 | 0.36 | 0.35 | 0.36 | 150 | 0.09 | -0.02 | -0.01 |
| wollongong01 | -0.45 | -0.19 | 0.63 | 0.89 | 0.78 | 0.91 | 157 | 0.26 | -0.26 | -0.13 |
| lauder02 | -1.00 | -0.77 | 0.55 | 0.56 | 1.14 | 0.95 | 104 | 0.23 | -0.01 | 0.19 |

Table S4: July statistics of $XCO_2$ [ppm] from 9km-EXP and 80km-EXP simulations with respect to TCCON stations. The differences between the two simulations (9km-EXP – 80km-EXP) are shown in the last three columns. The location of the station and their associated DOI are provided in Tab A3.

| Station | Bias 9kmEXP | Bias 80kmEXP | STDE 9kmEXP | STDE 80kmEXP | RMSE 9kmEXP | RMSE 80kmEXP | N data | $\triangle$ STDE | $\triangle$ \|Bias\| | $\triangle$ RMSE |
|---|---|---|---|---|---|---|---|---|---|---|
| bialystok01 | 0.97 | 1.01 | 1.06 | 1.17 | 1.44 | 1.54 | 68 | -0.04 | -0.11 | -0.1 |
| bremen01 | 1.18 | 1.36 | 0.67 | 0.77 | 1.36 | 1.57 | 44 | -0.18 | -0.10 | -0.21 |
| karlsruhe01 | 1.25 | 1.44 | 0.77 | 0.81 | 1.47 | 1.65 | 90 | -0.19 | -0.04 | -0.18 |
| orleans01 | 1.38 | 1.36 | 0.66 | 0.66 | 1.53 | 1.51 | 16 | 0.02 | 0.00 | 0.02 |
| garmisch01 | 0.92 | 1.03 | 0.66 | 0.74 | 1.14 | 1.27 | 90 | -0.11 | -0.08 | -0.13 |
| parkfalls01 | 0.56 | 0.49 | 1.09 | 1.07 | 1.23 | 1.18 | 168 | 0.07 | 0.02 | 0.05 |
| rikubetsu01 | 2.21 | 2.92 | 0.55 | 0.64 | 2.27 | 2.99 | 9 | -0.71 | -0.09 | -0.72 |
| lamont01 | 2.00 | 1.48 | 1.32 | 1.25 | 2.40 | 1.94 | 299 | 0.52 | 0.07 | 0.46 |
| tsukuba02 | 1.21 | 1.52 | 0.95 | 1.06 | 1.54 | 1.85 | 120 | -0.31 | -0.11 | -0.31 |
| edwards01 | 1.28 | 1.40 | 0.76 | 0.79 | 1.49 | 1.61 | 316 | -0.12 | -0.03 | -0.12 |
| pasadena01 | 1.23 | 0.57 | 1.30 | 1.09 | 1.79 | 1.23 | 302 | 0.66 | 0.21 | 0.56 |
| saga01 | 0.80 | 1.20 | 0.57 | 0.66 | 0.98 | 1.37 | 30 | -0.40 | -0.09 | -0.39 |
| izana01 | 0.45 | 0.11 | 0.36 | 0.49 | 0.58 | 0.50 | 43 | 0.34 | -0.13 | 0.08 |
| ascension01 | 0.08 | 0.18 | 0.45 | 0.47 | 0.46 | 0.51 | 158 | -0.10 | -0.02 | -0.05 |
| darwin01 | 1.54 | 1.55 | 0.33 | 0.35 | 1.58 | 1.59 | 264 | -0.01 | -0.02 | -0.01 |
| reunion01 | 0.81 | 0.79 | 0.33 | 0.36 | 0.87 | 0.87 | 136 | 0.02 | -0.03 | 0.0 |
| wollongong01 | 0.23 | 0.22 | 0.64 | 0.69 | 0.68 | 0.73 | 96 | 0.01 | -0.05 | -0.05 |
| lauder02 | 0.10 | 0.11 | 0.27 | 0.31 | 0.29 | 0.33 | 86 | -0.01 | -0.04 | -0.04 |